



# The Urmia Playa as source of airborne dust and ice nucleating particles – Part 1: Correlation between soils and airborne samples

Nikou Hamzehpour[1,2*], Claudia Marcolli[2], Sara Pashai[1], Kristian Klumpp[2], Thomas Peter[2]

[1]Department of Soil Science and Engineering, Faculty of Agriculture, University of Maragheh, Maragheh, Postal Box: 83111-55181, Iran.

[2] Department of Environmental Systems Science, Institute for Atmospheric and Climate Sciences, ETH Zurich, 8092 Zurich, Switzerland.

*Correspondence to*: Nikou Hamzehpour (nhamzehpour@maragheh.ac.ir)

**Abstract**

The emergence of desiccated lakebed sediments and their exposure to wind erosion as a consequence of climate change and drought in arid and semiarid regions of the world poses a growing hazard. Airborne dust originating from such soils can create health and environmental issues due to their high salt content and the presence of toxic elements. The aim of the present study is twofold, namely to investigate the newly emerged playa surfaces of the western Lake Urmia (LU) in Iran and their contribution to aerosol in the region by means of physicochemical, mineralogical and elemental analyses, and to study the ice nucleation (IN) activity of both surface-collected soil and airborne dust samples. The playa surfaces created by desiccation of LU on the west shores were mapped and sampled at 130 locations. Soil samples were subjected to physicochemical analyses and their erodible fraction was determined. Based on these analyses, four highly erodible playa surfaces from northwest to the south of LU were selected as sites for collection of dust by impaction and soil samples from the uppermost surface. Their particle physicochemical properties (size distribution, elemental and mineralogical composition) were compared with their IN activity determined by emulsion freezing experiments in a differential scanning calorimeter (DSC) in two suspension concentrations of 2 wt % and 5 wt %. The physicochemical soil properties differed significantly between the different playa surfaces, which affects their susceptibility to wind erosion. Sand sheets and sandy salt crusts were the most erodible playa surfaces due to their high sand fraction, low organic matter and clay content, favoring the presence of small aggregates. Mineralogical analyses document the prevalence of quartz, carbonates and clay minerals such as kaolinite, palygorskite and chlorite in all of the samples. The predominant elements in the samples are Ca, Fe, Al, Si and Na and in some cases Ba, Sr and Zn. The correlation between soil and dust samples based on mineralogical composition, elemental enrichment factors, and physicochemical properties confirm that the playa surfaces are the major contributors to dust in the region. IN activity with onset temperatures ranging from 245 to 250 K demonstrates the high potential of dust blown from Urmia playa surfaces to affect cloud properties and precipitation. Freezing onset temperatures and the fraction of heterogeneously frozen droplets in the emulsions reveal variations in IN activity depending on the mineralogical composition of the samples, but also influenced by organic matter, salinity and pH. Specifically, IN activity correlates positively with organic matter and clay minerals, and negatively with pH and salinity, and, surprisingly, also negatively with K-feldspar and quartz content. The high wind erodibility and dust production of the LU playa surfaces together with their high IN activity can play an important role in the climate of the region and thus needs careful monitoring and specific attention.

**Key words:** Elemental analysis, enrichment factor, ice nucleation, mineralogy, playa surfaces, wind erodibility



## 1 Introduction

Wind erosion is one of the main soil degradation processes in arid and semiarid regions, which generates dust carrying millions of tons of soil particles into the atmosphere (Middleton, 2017). Dust phenomena are an important environmental problem in different regions, putting at risk both human health and ecosystem safety (Goudarzi et al., 2019; Kim et al., 2017). Depending on their size distribution, dust from deserts and arid or semiarid regions of the world can be transported over long distances and even between continents (Middleton and Goudie, 2001; Prospero et al., 2002; Moreno et al. 2006; Schepanski, 2018; Froyd et al., 2022).

Saline dust storms mostly originate from dried lakebeds and saline soils on the margins of lake floors (Abuduwaili et al., 2010). Saline dust storms are different from common dust storms in terms of the sources of suspended PM, chemical composition, and grain size distribution. It has been shown that enormous amounts of dust are deflated annually from the Aral Sea in Kazakhstan and Uzbekistan and Ebi Nur Lake in northwestern China (Abuduwaili et al., 2010).

In general, specific hydrologic conditions, soil properties and physicochemical composition are required to form a specific playa surface (Krinsley, 1970), and are also major parameters in controlling dust generation by each surface. Therefore, having knowledge about the different types of playa surfaces and their physicochemical characteristics help to identify dust prone surfaces and could be a step forward in their control. In this regard, several studies have analyzed playa geomorphic surfaces and their potential in dust generation (Reynolds et al., 2007; Quick et al., 2009; Halleaux and Rennó, 2014; Bowen and Johnson, 2015; Wurtsbaugh et al., 2017). For example, Wurtsbaugh et al. (2017) have shown that the release of particles from the dried bed of Owens Lake in California increased the $PM_{10}$ levels and caused respiratory problems. Bowen and Johnson (2015) found that the increase in soil humidity decreases the likelihood of dust generation on a playa surface. Reynolds et al. (2007; 2009) found that dry playas of Mojavi desert were prone to dust generation following wet periods.

Dust storms in Iran used to originate from the neighboring countries, such as Iraq and Syria (Zarasvandi, 2009; Moridnejad et al., 2015). Yet, during the past decades, a number of lakes are shrinking and in danger of drying (e.g. Hamon, Bakhtegan, Gavkhoni), thus creating new dust sources all over the country. Therefore, a number of studies have focused on the susceptibility of dried lakebeds to wind erosion (e.g. Farpoor et al., 2012; Ghadimi and Ghomi, 2013; Rashki et al., 2013a; Shahryary, 2014). When Raisossadat et al. (2012) studied the geomorphology and genesis of Sahlabad Playa in the east of Iran, they found two types of clay and salt plains along with mud flats with several faces including puffy and soft faces, plough surfaces, clay plains, Nabka areas, and clay-slate polygons. Farpoor et al. (2012) identified several geomorphic surfaces in Sirjan Playa including salty and non-salty clay flats (CFs), puffy grounds (PG), salt crusts (SCs) and wet zones. It has also been shown that desiccation of Hamoun Lake in southeast Iran has increased the frequency and intensity of dust storms (Rashki et al., 2013a; 2013b). Ghadimi and Ghomi (2013) found that Mighan Playa in Arak city, central province of Iran, was ephemeral in all phases and very unstable.

The surface area of Lake Urmia (LU) in the northwest of Iran (Fig. 1) has been declining during the past few decades. It has shrunk almost to half of its maximum extension (6000 km² in 1998) by 2021, leaving behind vast barren lands with high salt content and low vegetation cover in most parts (Kakahaji et al., 2013; Farokhnia and Morid, 2014; Shadkam et al., 2016; Hamzehpour et al., 2018).

This playa type, saline areas with limited vegetation cover in most parts (see Fig. 2 for examples), might be of high vulnerability to wind erosion. Studies have also shown that the intensity of dust storms over the LU and nearby cities have increased (Gholampour et al., 2015; Sotoudeheian et al., 2016; Boroughani et al., 2019; Ahmady-Birgani et al., 2020). Ahmady-Birgani et al. (2020) demonstrated that areas impacted by aerosols emitted from LU bed can extend to 40 km from its shoreline. Saline





aerosols and crustal particles from LU dried beds account for about 60 % of $PM_{10}$ in the region, reaching values about nine times higher than the WHO annual guideline value (20 µg m$^{-3}$) (Gholampour et al., 2015; 2017).

Alkhayer et al. (2019) studied Lake Urmia Playa (LUP) and prepared reconnaissance maps of areas susceptible to wind erosion from northeast to the southwest of LU. The results showed that approximately 35 % of the surfaces are strongly susceptible to wind erosion and highly prone to generate saline dust and sand storms. These areas were located in the east and southeast of the lake. Also, 40 % of the playa surfaces had moderate resistance to wind erosion and can become very sensitive to wind erosion if ground water depth or the roughness of the surface changes. In a detailed study in the southeastern part of LUP, Motaghi et al. (2020) reported that the playa surface type is a determining factor for its resilience to wind erosion. They found that soil erodible fraction varied between 40 % and 90 % among playa surfaces. Salt crusts and clay flats were more resilient against wind erosion while salt crusts-clay flats, agricultural lands and abandoned agricultural lands had the highest erodible fractions.

 Several studies have revealed that geochemical composition analysis of dust particles in the deposit area can be used to identify the potential sources of dust generation (Reheis et al., 2002; Derbez and Lefèvre, 2003; Abouchami et al., 2013; Zarasvandi et al., 2011; Zhang et al., 2017). Among limited works in this regard in LU basin, Gholampour et al. (2017) studied the elemental composition of particulate matter in the southeastern and northern parts of Lake Urmia. They found that NaCl, $BaCl_2$ and the elements Al, Ti, Ca, P, Mn, K, F and Si mainly originate from LU crustal soil. However, the lack of air pollution monitoring stations around LU has limited information about the direct contribution of LU in dust events over the region.

Airborne dust is also relevant as source of ice nucleating particles (INPs) (Froyd et al., 2022), which are required for primary ice production in mixed phase clouds, and, ultimately, for precipitation formation over land in midlatitudes (Mülmenstädt et al., 2015). Mineral dust particles have been found to be the dominant INP type at temperatures below 258 K (Hoose and Möhler, 2012; Murray et al., 2012). Indeed, Brunner et al. (2021) estimated that 97 % of all INPs active at 243 K at the High Altitude Station Jungfraujoch, Switzerland, are dust particles. At higher temperatures, biological particles are considered more relevant (Conen et al., 2011; Kanji et al., 2017; Testa et al., 2021). Whether dust particles act as potent INPs depends on their mineralogical composition and mixing with other aerosol components (Murray et al., 2012; Kanji et al., 2017). Feldspars, and especially K-feldspars proved to nucleate ice at the highest temperatures among the most common mineral dust species, and have therefore been considered the most important minerals for cloud glaciation (Atkinson et al., 2013). Yet, its ice nucleation (IN) activity proved to be highly sensitive to the presence of other aerosol constituents and to decline in the presence of ions or acids (Kumar et al., 2018; Whale et al., 2018: Yun, et al., 2020; 2021; Klumpp et al., 2022). Freezing experiments with quartz particles also yielded mixed results. Grinding increased its IN activity, which deteriorated again after aging in water over longer periods of time (Zolles et al., 2015; Kumar et al., 2019a). Boose et al. (2016) found a correlation between quartz content and IN activity for desert dusts, yet, the samples responsible for this positive correlation had been ground before freezing experiments were performed. Hence, the role of quartz as INP in natural dust samples is uncertain. Clay minerals such as illite (Hiranuma et al., 2015), kaolinite and montmorillonite all showed IN activity in lab experiments, but typically at lower temperatures than ground quartz and K-feldspars (Pinti et al., 2012; Hoose and Möhler, 2012; Hiranuma et al., 2015; Kanji et al., 2017). But also for a specific mineral type, IN activity is not identical for all particles, but shows a distribution with some, mostly smaller particles being completely inactive and larger ones inducing freezing at higher temperatures (Marcolli et al., 2007: Lüönd et al., 2010; Welti et al., 2019). Fertile soil dust has been found to nucleate ice at higher temperatures than desert dust, which has been attributed to biological and/or organic content e.g. through testing the heat resistance of their ability to nucleate ice (Conen et al., 2011; Tobo et al., 2013; 2014; O'Sullivan et al., 2014; Hill et al., 2016). Only few studies (O'Sullivan et al., 2014; Kaufmann et al., 2016; Boose et al., 2016; Paramonov et





al., 2018) have related the IN activity of natural dust samples to their mineralogical composition. Yet, parameterizations of atmospheric INP concentrations depending on dust source regions would require information about their size-resolved mineral composition (Perlwitz et al., 2015a; 2015b).

Here we identify dust sources around LU depending on soil type and erodible fraction. For four highly erodible source regions, we determined the physicochemical properties like salinity, pH, and particle size distribution, and analyzed the mineralogical and
elemental composition of (surface collected) soil samples and compare them with (air collected) dust samples from nearby meteorological stations. Finally, we measured the IN activity of the samples and correlated the heterogeneous freezing signal with the composition of the samples.

The manuscript is structured as follows: Sect. 2 presents the materials and methods applied in this study. Results and Discussion (Sect. 3) start with the assessment of the erodibility of the western LUP surfaces and their characterization based on samples
collected at 130 locations (Sect. 3.1). With these findings, four highly erodible locations were identified to collect surface soil samples together with airborne dust samples from nearby meteorological stations (Sect. 3.2). These samples were analyzed and correlated with respect to mineralogical composition (Sect. 3.2.1), elemental composition (Sect. 3.2.2), and physicochemical properties (Sect. 3.2.3). Sect. 3.3 compares the IN activity of the soil and dust samples with each other and with mineralogical composition and physicochemical properties, followed by the conclusions in Sect. 4.


## 2 Materials and Methods

### 2.1 Lake Urmia Playa surfaces and their wind erodibility

#### 2.1.1 Delineation and mapping of the playa surfaces

LUP surfaces from the northwest to the south of Lake Urmia, which have been developing as a consequence of the gradual recession
of the lake were studied during spring to summer 2018. The study area is located between 45º 06' to 45º 23' E and 37º 19' to 37º 56' N and is adjacent to highly productive agricultural farmlands. The region has a mean annual precipitation of 367 mm, a mean annual temperature of 13.4°C, and potential evaporation value of 900–1170 mm (Iran Ministry of Energy, 2014).

Using Landsat images acquired for the study period, different playa surfaces were distinguished and demarcated based on the variations in color reflectance of the images and vegetation cover. Through extensive fieldwork, each demarcated map unit was
checked and named in the field based on the concept of playa surfaces defined by Krinsley (1970) for Iranian playas. Different playa map units were distinguished based on major variations in their groundwater table, vegetation cover type and density, type of soil crusts, and their rupture resistance (Table 1). Then, from all playa surfaces based on their surface area and variations inside map units, soil samples were taken from the 0–5 cm surface layers. Over all 130 soil samples were collected and subjected to physicochemical analysis. Finally, a map of the playa surfaces was prepared in ArcGIS 10.1 software (Fig. 3).

#### 2.1.2 Physicochemical analysis of the collected soil and dust samples

The collected soil samples from playa surfaces were passed through a 2 mm sieve and the following physicochemical properties were determined: Soil electrical conductivity (EC) and acidity (pH) were measured in 1:2.5 soil to water extracts using a Jenway conductivity meter (model 4510) and VWR Symphony SB70P pH meter, respectively (Rhoades, 1996).



In addition, soil organic carbon (OC) was measured using a wet oxidation technique (Nelson and Sommers, 1996); and soil total
carbonates were determined using a back titration of the remaining HCl (Page et al., 1982). Organic carbon was transformed to
organic matter (OM) by multiplying by a factor of 2.

Soluble ions ($Na^+$, $Ca^{2+}$ + $Mg^{2+}$, $Cl^-$) were also quantified in a 1:2.5 soil to water extract. $Na^+$ was measured using a Sherwood
Flame photometer (model 410) and total ($Ca^{2+}$ + $Mg^{2+}$) was determined using a titration with EDTA; while $Cl^-$ was determined by
titration with 0.02 N silver nitrate solution (Sparks et al., 2020).

Soil texture was determined using a hydrometer method (Gee and Or, 2002). A rotary sieve in dry state according to Nimmo and
Perkins (2002) was used to determine the fraction of soil materials < 0.84 mm as an indicator of wind erodible fraction (EF) using
the following equation:

$$EF = \frac{W < 0.84}{TW} \times 100 \tag{1}$$

where EF is soil erodible fraction in (%), W < 0.84 is weight (g) of soil material smaller than 0.84 mm and TW is the initial weight
(g) of the total sample after air drying. EF is an indicator of the susceptibility of soils to wind erosion as it shows which portion of
a soil has the potential to be eroded by wind.

In addition, mean weight diameter (MWD) was determined in unsieved samples by dry sieving using 0.25, 0.5, 1, 2, and 4.75 mm
sieve series (Kemper and Chepil, 1965). MWD was calculated as (Van Bavel, 1950):

$$MWD = \sum_{i=1}^{n} x_i y_i \tag{2}$$

where $x_i$ is the mean diameter of the size classes in mm and $y_i$ is the proportion of each size class by weight with respect to the
total sample.

In the collected soil samples from highly erodible playa surfaces (< 2 mm) and dust samples (see Sect. 2.2.2 for dust sample
collection) from nearby areas (unsieved samples), soil EC, pH, OC were also determined following the procedures discussed above.
In addition, as the size distribution of particles (PSD) is important for the potential of a soil to remain airborne and be transported
over long distances, a Laser Diffraction Size Analyzer (LDSA; model LS 13320) was used to determine PSD between 0.4 µm to
2000 µm for the collected air-dried soil and dust samples. PSD analyses were also performed for the 63 µm sieved fraction. The <
63 µm fraction was then used for ice nucleation measurements. In order to disperse the samples, they were wet sonicated for 5 min
prior to analysis (Dane and Topp, 2020).

### 2.1.3 Statistical analysis of soil erodible fraction over different playa surfaces

A Completely Randomized Design (CRD) was applied to playa surfaces as treatments and unequal replications to investigate how
the type of playa surfaces affects their wind erodible fraction. A similar method was also used to compare the physicochemical
properties of different playa surfaces. The experimental data were analyzed using the SASS software (version 9.1), and the mean
comparison was conducted using the Duncan's Multiple Range Test (DMRT) (P < 0.05).

### 2.2 Dust collection and source identification

**2.2.1 Meteorological data**

Maximum wind speed and wind direction data at the weather stations closest to the four sampling locations (Table 4) for the dry months (June – October 2020), which include the sampling month (July 2020), were obtained from the National Climatic Center website (https://data.irimo.ir/). The retrieved data was examined for missing values and outliers to input in WRPLOT View Freeware 8.0.2 to plot the wind rose.

**2.2.2 Dust and soil sample collected from erodible surfaces**

Soil samples from highly erodible surfaces next to the LU were collected at four locations together with dust samples from nearby meteorological stations as presented in Fig. 1, (yellow: soil sampling locations; red: dust sampling locations). The soil samples Jabal (Jab) and Merange (Mer) were collected from two highly erodible playa surfaces identified in the western part of the LUP, namely sa-sheets and Sa-SC (salt-crusts) (Figs. 2b and 2c). The other two dust sampling locations were chosen according to the

identification of major wind erodible surfaces of LUP from previous studies (Alkhayer et al., 2019; Motaghi et al., 2020). The one from nearby abandoned agricultural lands in the northern part of the study area is named Salmas (Sa) (Fig. 2a), and the other one from salt crusts-clay flats in the southern part of the study area is named Miandoab (MD) (Fig. 2d).

At each location soil samples were collected from the top 5 cm surface layer. The dust samples were sampled around 3 m above ground using high-volume samplers manufactured by Graseby–Andersen (Smyrna, Georgia, USA) on a 20.3 cm × 25.4 cm glass

micro-fiber filter (Whatman Inc., USA) at flow rates of 1.13–1.41 $m^3 min^{-1}$ for 24 h.

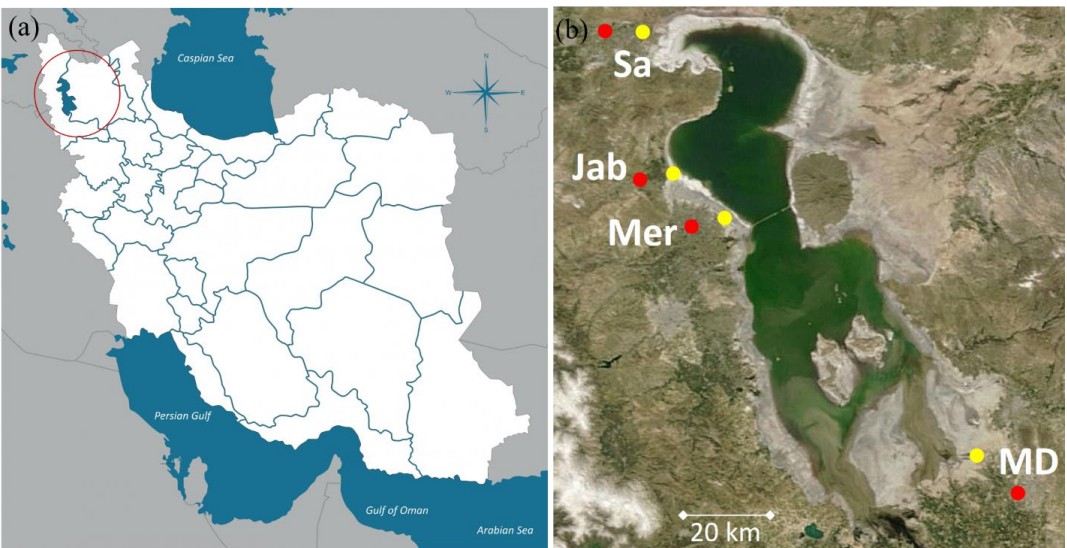

**Figure 1:** (a) The location of the study area in the northwest of Iran, shown with red circle. (b) Sampling locations from the northwest to the southeast of Lake Urmia. Red and yellow circles correspond to the dust and soil sampling locations, respectively. Sa: Salmas; Jab: Jabal; Mer: Merange; MD: Miandoab. Lake Urmia image taken on 23 April 2016 by MODIS on NASA's Aqua satellite. Courtesy of NASA's Earth

Observatory.

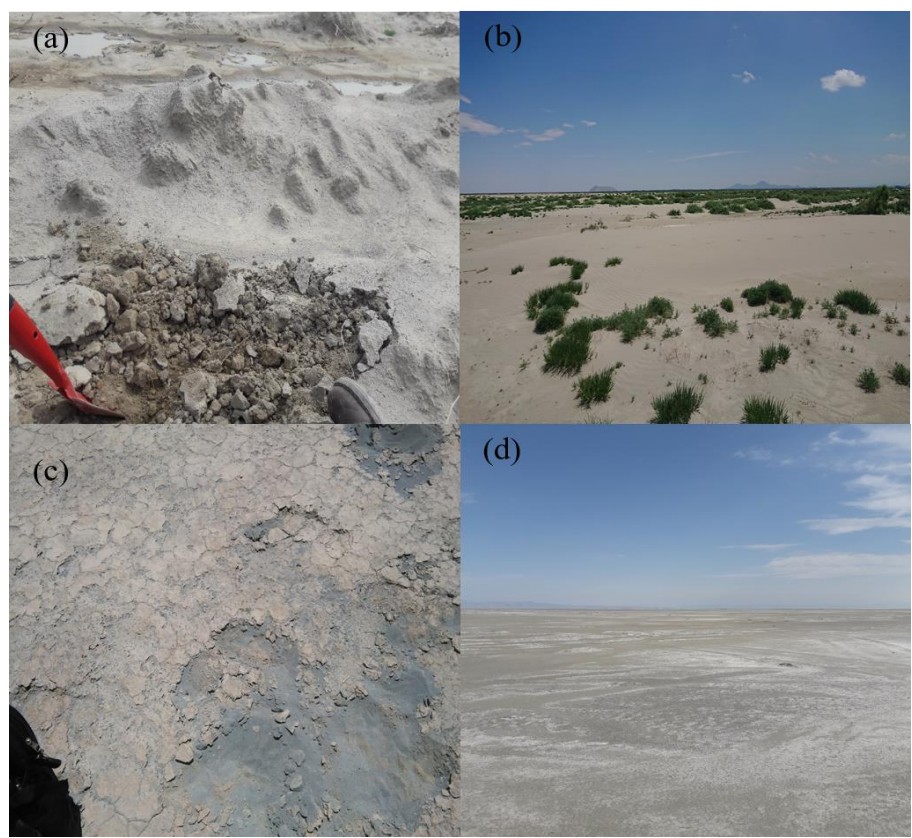

**Figure 2:** Images of LUP surfaces with high susceptibility to wind erosion, where the samples were collected. (a) abandoned agricultural lands (Salmas soil sample); (b) sand sheets (Jabal soil sample); (c) sandy salt crusts (Merange soil sample); (d) salt crusts-clay flats (Miandoab soil sample).

### 2.2.3 Quantitative mineralogy

Mineralogy of the dust and the sieved (< 2 mm) soil samples was determined with X-ray diffraction (XRD) analysis (Bish and Plötze, 2011). An aliquot of about 0.5 g was milled in ethanol to a grain size below 20 µm with a McCrone micronizing mill and dried afterwards at 65 °C. For randomly oriented powder specimens frontloading preparation was carried out. The powdered material was gently pressed in a sample holder for packing, sample-height adjustment and forming a flat surface. Preferred orientation was minimized using a blade (Zhang et al., 2003). A second sample preparation was carried out producing oriented specimens for enhancement of the basal reflexes of layer silicates thereby facilitating their identification. The changes in the reflex positions in the XRD pattern by intercalation of different organic compounds (e.g. ethylene glycol) and after heating were applied.

X-ray diffraction measurements were made using a Bragg-Brentano X-ray diffractometer (D8 Advance, Bruker AXS, Germany) using CoKα (35 kV, 40 mA) radiation. The instrument was equipped with an automatic beam optimization setup (theta compensating divergence slit, automatic air scatter screen and a Lynx-Eye XE-T detector). The powder samples were step-scanned at room temperature from 2 to 80°2Theta (step width 0.02°2Theta, counting time 2 s per step). The qualitative phase analysis was





carried out with the software package DIFFRACplus (Bruker AXS). The phases were identified based on the peak positions and relative intensities in comparison to the PDF-2 database (International Centre for Diffraction Data).

The quantitative amount of the mineral phases was determined by means of Rietveld analysis of the XRD diffractograms. This full

pattern-fitting method consists in the calculation of the X-ray diffraction pattern and its iterative adjustment to the measured diffractogram. In the refinements phase specific parameters and the phase content were adapted to minimize the difference between the calculated and the measured X-ray diffractograms. The quantitative phase analysis was carried out with Rietveld program Profex/BGMN (Döbelin and Kleeberg, 2015).

### 2.2.4 Elemental analysis

Approximately 150 mg of each sample was weighed into a high-pressure Teflon digestion vessel. First, 6 mL of $HNO_3$ (70%) was added to the sample leading to reaction. When the reaction had lessened, 18 mL HCl (35%) were added and the vessels were closed for 1 h. After digestion, samples were transferred into a pre-cleaned 50 mL Greiner tube, topped up to 50 mL with deionized water ($\geq$ 18.2 M$\Omega$) and centrifuged for 5 min with 4000 rpm (approx. 2700 g). Then, approximately 10 mL of sample was transferred into a 15 mL pre-cleaned Greiner tube, which was used as storage vessel. The digested samples were diluted by 1:20 and 1:100

v/v in 1 % $HNO_3$ v/v prior to measurements.

The elements Li, Be, B, Na, Mg, Al, Si, P, K, Ca, S, V, Cr, Mn, Fe, Co, Ni, Cu, Zn, Ga, As, Se, Rb, Sr, Mo, Cd, Cs, Ba, Ti, Pb, Bi, U were quantified by inductivity coupled Plasma Mass Spectrometry (Agilent 8900 QQQ ICP-MS (HMI mode 4). Measurement detection limits (MDLs) were determined as three times of the standard deviations of the blank values (five replicates of the blank). Efficiency of recovery was determined by measuring known amounts of elements. Detection limits, MDLs, and

recoveries are presented in Table S2 in the Supplementary Materials.

### 2.2.5 Enrichment Factor

Aerosols may be related to their emission source in terms of size, physicochemical properties and elemental composition. To investigate the relevance of LUP sediments for dust loadings in nearby areas, it is important to differentiate between particles emitted from earth crust and those from LUP sediments. Enrichment factors allow establishing the probable origin of elements in

PM as proposed by Zoller et al. (1974). They are calculated as the ratio of chemical concentration of an element in the aerosol sample to that in the soil sample:

$$\text{Enrichment factor} = \frac{(C_x/C_{\text{Fe}})_{\text{PM}}}{(C_x/C_{\text{Fe}})_{\text{crust}}} \tag{3}$$

Where $C_x$ is the concentration of element X and $C_{Fe}$ is the concentration of Fe as reference (or the element with the highest concentration in the soil crust).

Enrichment factors less than one identify the local soil crust as the main source of an element; enrichment factors of 1–5 indicate

other emission sources besides the soil crust; enrichment factors > 5 imply other emission sources as the predominant sources; finally, enrichment factors > 10, suggest that the source of the element is mainly non-crustal.



### 2.3 Ice nucleation measurements of soil and dust samples

For the immersion freezing measurements, a differential scanning calorimeter (DSC) Q10 from TA instruments was used. Emulsion freezing experiments were performed to characterize the ice nucleation efficiency of the soil and dust samples. All samples were passed through a 63 µm sieve and examined at two concentration levels (2 wt % and 5 wt %) in aqueous suspensions (molecular bioreagent water, Sigma Aldrich). After preparing sample suspensions, they were sonicated for 5 min in order to minimize particle agglomeration. Then the suspensions were combined with a mixture of mineral oil and lanolin (both Sigma Aldrich) at a ratio of 1:4 and emulsified with a rotor stator homogenizer (Polytron PT 1300D with a PT-DA 1307/2EC dispersing aggregate) for 40 s at 7000 rpm. 5 to 10 mg of emulsion were placed in an aluminum pan, hermetically sealed and placed in the DSC. Cooling and heating cycles were run at a rate of 1 K/min in the temperature ranges for freezing and melting. These were used for evaluation of the freezing onset temperatures of the heterogeneous ($T_{het}$) and homogeneous ($T_{hom}$) freezing peaks, the heterogeneously frozen fraction ($F_{het}$) and the melting temperature $T_{melt}$ as explained in Kumar et al., (2018). To test the stability of the emulsions, some samples were subjected to three freezing cycles following the procedure introduced by Marcolli et al. (2007) with a first and third cycle performed at a cooling rate of 10 K/min as control cycles. Emulsions were freshly prepared before each experiment. Every experiment was repeated at least once with a freshly prepared suspension.

### 3 Results and Discussions

#### 3.1 Determining wind erodible playa surfaces for soil sampling and dust collection

#### 3.1.1 Urmia Playa Lake surfaces and their characteristics

In Table 1, the summary of the investigated playa surfaces along with their groundwater depth, type of surficial crusts, crust stability and vegetation cover density are presented. In Fig. 3, the map of prevailing playa surfaces and their distribution in the study area is shown.

Seven type of playa surfaces were observed in the study area: salt crust (SC), clay flat (CF), clay flat-salt crust (CF-SC), sand sheets (sa-sheets), sandy salt crust (Sa-SC), beach sand, and fan delta (FD). CFs have developed in the farthest distance from LU in the margin of the Urmia Plain in the west of LU (Fig. 3) due to the impact of precipitation and high clay content (physical development of soil crust). Based on their different soil crust thickness, crust rupture thickness, and vegetation cover density, CFs have been subdivided in eight map units. Overall, these lands are low in salinity and ground water depth is deeper than 4 m (Table 1). They have relatively stable crusts and the thickest crusts belong to this group. They have also dense vegetation cover in most parts making them less susceptible to wind erosion. CF-SCs in the region have developed in a transition zone between CFs and Sa-SCs, where due to elevation variation and as its consequence variable groundwater depth, CFs and SCs have co-evolved. They have been further subdivided into nine map units (CF-SC1 – CF-SC9). On average, their crust thickness and stability is lower than the one of CFs (Table 1). These surfaces have lower vegetation cover due to the higher soil salinity compared to CFs and along with physical crusts chemical crusts have developed over these playa surfaces (physical and chemical type of soil crust).





**Table 1**. Major field characteristics of distinct playa map units sorted by types of playa surfaces.

| Playa surface type | Number of samples | Map unit | Depth to ground water (m) | Type of soil crust | Soil crust thickness (cm) | Crust rupture resistance | Vegetation cover density (%) |
|---|---|---|---|---|---|---|---|
| Clay Flat (CF) | 37 | CF1 | > 4 | Physical | < 0.5 | W | 40–60 |
| | | CF2 | | | 2–3 | VS | 50–70 |
| | | CF3 | | | 2.5–3.5 | VS | 50–70 |
| | | CF4 | | | 2.5–3.5 | VS | 50–70 |
| | | CF5 | | | 3–5 | ES | 80–100 |
| | | CF6 | | | 0.5–1 | M | 40–60 |
| | | CF7 | | | 3.5–5 | VS | 40–60 |
| | | CF8 | | | 0.5–1 | M | 40–60 |
| Clay Flat-Salt Crust (CF-SC) | 50 | CF-SC1 | < 2 | Physical-Chemical | 1.5–2 | W–M | 10–30 |
| | | CF-SC2 | | | 1.5–2 | W–M | 10–30 |
| | | CF-SC3 | | | 1.5–2 | M–S | 20–30 |
| | | CF-SC4 | | | 1.5–2 | M–S | 10–20 |
| | | CF-SC5 | | | 3–5 | ES | 50–70 |
| | | CF-SC6 | | | 2.5–3.5 | VS | 60–70 |
| | | CF-SC7 | | | < 0.5 | W | 30–50 |
| | | CF-SC8 | | | 3–5 | ES | 50–70 |
| | | CF-SC9 | | | < 0.5 | W–VW | 0 |
| Sa-SC | 20 | Sa-SC | < 0.5 | Chemical | 1.5-2 | M | < 10 |
| Salt Crust | 13 | SC | < 0.1 | Chemical | 0.5 > 5 | M–ES | < 5 |
| Sa-sheets | 6 | Sa-sheets | > 2 | - | 0 | - | 10–30 |
| Beach sand | 4 | Beach sand | > 1 | - | 0 | - | 70–90 |

SC: salt crust; CF: clay flat; ES: extremely strong; VS: very strong; VW: very weak; EW: extremely weak; M: moderate.

Both Sa-SCs and SCs have shallow groundwater depth with chemical crusts (salt accumulation and deposition on the top of the soil). The major difference leading to categorize them in two different playa surface types is the higher sand fraction of Sa-SCs compared with SCs. The development of both soil types is dominated by chemical drivers. Sa-SCs mostly developed in areas
where LU expands during pluvial years while SCs have developed close to the LU brine pool where the soil is most of the year covered with a thin layer of water.



Sa-sheets occur in a limited area in the northern part of the study area (Fig. 3) with almost no vegetation cover nor surficial crusts due to their sandy texture. However, in some parts of this region artificial vegetation covers and also dead plant residues have been established to control aerosol lift. Overall, this area is highly susceptible to wind erosion. Fan deltas (FDs) formed where rivers

from the LU basin enter the playa sediments at the mouth of the river. Due to the fine texture of playa sediments and the low relief, rivers divide into several branches. No soil samples were taken from these locations due to waterlogging. Beach sands were observed in a very small area. They exhibit dense vegetation cover with deep roots and although this region is composed of sand, it is completely stabilized by vegetation cover.

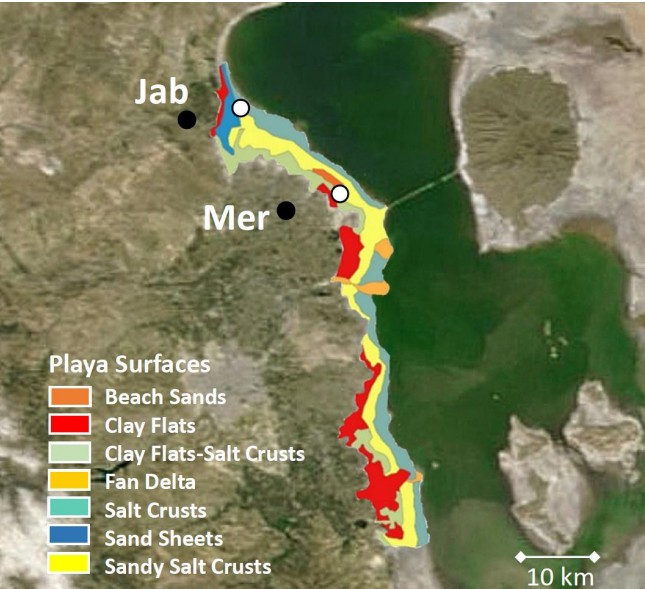

**Figure 3:** Map of playa surfaces in the western part of Lake Urmia investigated in this study. The soil and dust sampling sites (black and white circles, respectively) at Jabal (Jab) and Merange (Mer) are also shown. Lake Urmia image taken on 23 April 2016 by MODIS on NASA's Aqua satellite. Courtesy of NASA's Earth Observatory.

### 3.1.2 Physicochemical properties of Lake Urmia Playa soil types

Following the categorization of the playa surface, the physicochemical properties of the different soil types were determined as summarized in Table 2 and visualized in Fig. 4. Electrical conductivity is a standard measure to infer soil salinity, as it is sensitive to the ion content of soils, though, it is also influenced by clay content and organic matter (Corwin and Lesch, 2005). Values above 10 dS m$^{-1}$ represent high salinity, while values below 2.5 dS m$^{-1}$ stand for very low salinity (Rhoades, 1996). Thus, all soil types with the exception of beach sand exhibit very high to extremely high salinity. The most saline samples with a mean EC value of

117.7 dS m$^{-1}$ originate from SC sediments close to the current LU brine pool. Their high salt content is due to the increased water salinity of LU following its recession during dry climatic conditions. The main cations and anions responsible for the high salinity are $Na^+$, $Mg^{2+}$, $Ca^{2+}$ along with $Cl^-$. Therefore, the cation concentration especially [Na] is paralleled by EC (Fig. 4a). Liu et al. (2011) found that in the topmost 2 cm of salt crusts, $Na^+$, $Ca^{2+}$ and $Mg^{2+}$ are the main cations with salt contents that can exceed 40 %.





The second highest EC and cation concentrations were measured for the sandy salt crusts (Sa-SC), which developed next to the SCs in a farther distance from the LU brine pool. The lowest EC and cation concentrations were observed in beach sand, which seems to have been part of the LU shore at its time of largest expansion (Fig. 3). Note that the conductivity of the sand sheets stems from $K^+$ rather than $Na^+$ ions.

**Table 2.** Mean physicochemical properties of the different types of identified playa surfaces.

| Playa surface type | EC (dS m$^{-1}$) | pH | Ion conc. (cmol L$^{-1}$) | | OM (%) | TC (%) | Size fractions (%) | | | | EF (%) | MWD (mm) |
|---|---|---|---|---|---|---|---|---|---|---|---|---|
| | | | Ca$^{2+}$+ Mg$^{2+}$ | Na$^+$ | | | Clay | Silt | Total Sand | Fine Sand | | |
| CF | 26.5 | 8.1 | 15.8 | 50.9 | 2.9 | 17.7 | 36 | 37 | 27 | 10 | 25.0 | 2.1 |
| CF-SC | 43.6 | 8.2 | 19.6 | 65.9 | 1.6 | 21.4 | 27 | 41 | 32 | 15 | 36.3 | 1.9 |
| Sa-SC | 84.6 | 8.3 | 28.2 | 118.0 | 1.3 | 23.6 | 19 | 27 | 54 | 20 | 42.6 | 1.9 |
| SC | 117.7 | 8.7 | 27.2 | 146.7 | 1.9 | 17.9 | 31 | 45 | 24 | 13 | 22.5 | 2.4 |
| Sa-sheets | 30.1 | 8.1 | 1.1 | 32.6 | 0.5 | 17.4 | 12 | 5 | 83 | 44 | 89.7 | 0.7 |
| Beach sand | 0.4 | 8.3 | 1.7 | 2.3 | 6.0 | 38.0 | 12 | 12 | 76 | 3 | 55.0 | 1.7 |

EC: electrical conductivity; OM: organic matter; TC: total carbonates; clay: < 2 µm; silt: 2–50 µm; sand: 50–2000 µm; fine sand: 50–200 µm; EF: erodible fraction; MWD: mean weight diameter. CF: clay flats; SC: salt crusts; Sa-SC: sandy salt crusts.

Soil organic matter is only present in relatively low amounts and varies between 0.5 % and 6.0 % (Table 2). These are typical
values for soils in the western LU region, where OM is mostly below 3 % (Hamzehpour et al., 2018). For comparison, soil OM for annual crop fields in Iran varies from 0.34 % to 4.88 % with an average of 2.1 % (Bahadori and Tofighi, 2016). The lowest OM was measured for soil samples from sand sheets of the northern part of the study area (Fig. 3) with very low vegetation cover along with high sand fraction (Table 2). For beach sand playa surfaces, which exhibit dense vegetation cover, the highest OM values (6.0 %) were measured (Fig. 4c). CFs have the second highest OM content due to a considerable, salt-resistant native
vegetation cover with dense and extended roots. Moreover, due to their high soil clay content their water holding capacity is high, which also may explain their high OM content.

In some SC map units, OM values above average were determined although almost no detectible vegetation cover existed. This could be due to the transport of OM by rivers to the LU and their deposition during LU recession. Waterlogging conditions in these areas due to the shallow ground water table together with high soil salinity could have led to very slow soil organic matter
decomposition and therefore its accumulation.

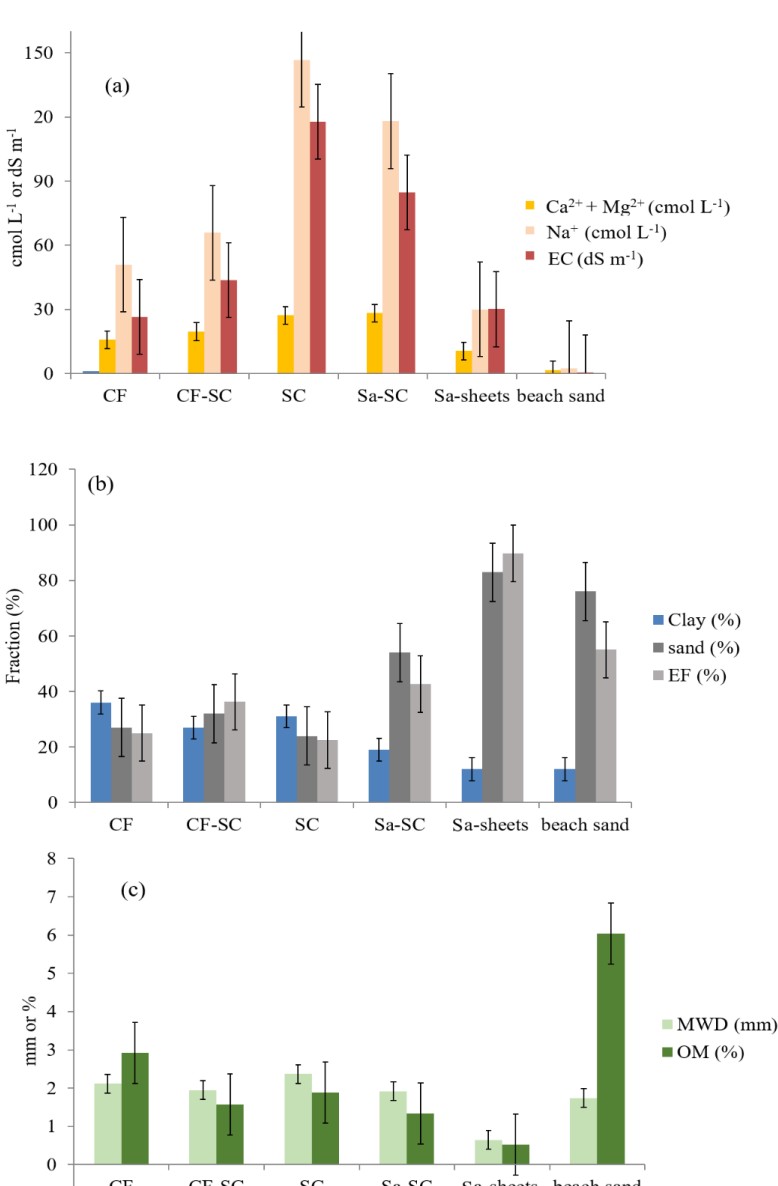

**Figure 4:** Comparison of the mean physicochemical properties of soil samples among different playa surfaces. Panel (a) compares $Ca^{2+} + Mg^{2+}$, $Na^+$, and EC values over different playa surface types; (b) comparison of clay, sand, and EF values over different playa surface types; (c) comparison of MWD and OM over different playa surface types. Bars represent the error bar.

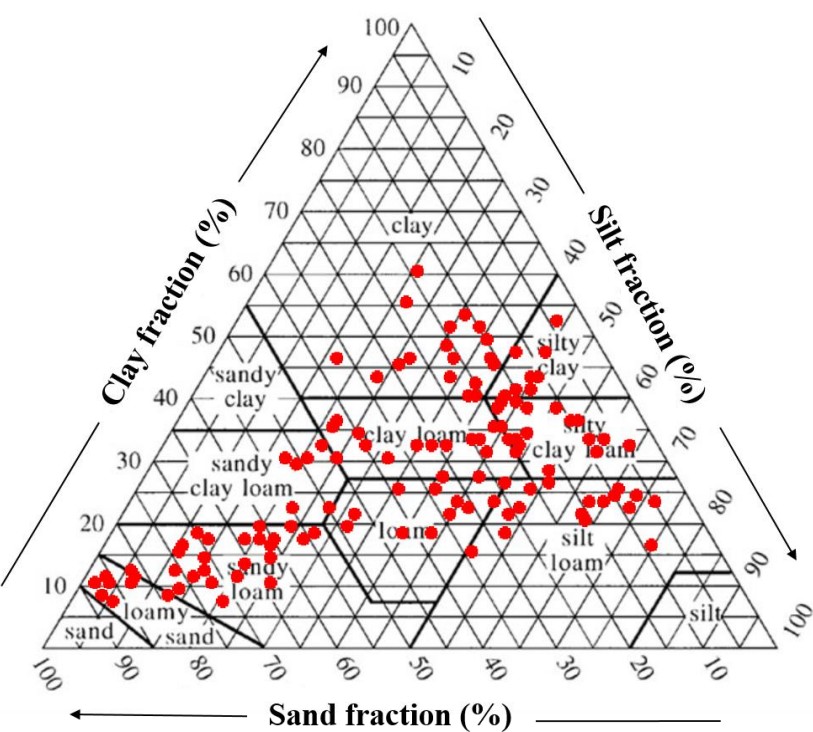

**Figure 5:** The location of the studied soil samples on the soil textural triangle. Created online at www.nrcs.usda.gov .

The clay fraction of the soils varies between 12 and 36 % between the playa surface types (Table 2). The highest amounts of clay were observed in CFs (Figs. 4b), where the finest sediments are deposited due to the reduced river carrying capacity in playa margins. Clay fraction was lowest in soil samples taken from the sand beach and sand sheets of LUP. The sand fraction is anticorrelated with silt and clay, with highest concentrations in sand sheets and sandy salt crusts (Fig. 4b). Sand-sized particles had been deposited on these playa surfaces while they had been part of the LU beach.

The Mean Weight Diameter (MWD) obtained by dry sieving is an indicator of aggregate stability in dry soils. Smallest MWD were measured in soil samples from sand sheets in the northern part of the study area (Figs. 3 and 4c), where the absence of vegetation cover along with very low clay fraction results in smaller-sized aggregates (Boruvka et al., 2002), and makes the soil susceptible to wind erosion. Conversely, the highest MWD values were measured for CFs and SCs and can be explained by the high clay and OM content in the case of CFs, and the high salt content in SCs, which can act as a cementing agent (Farid Giglo, 2014).

Overall, there is a high diversity of soil textures present in the studied samples, with sandy loam, clay loam and clay as the prevailing classes (see Fig. 5).





### 3.1.3 Correlation between soil properties of playa surfaces and their wind erodibility

Soil erodible fraction (EF) is a determining factor for soil susceptibility to wind erosion. The Pearson correlation coefficient between soil erodible fraction and soil properties demonstrates negative correlations to soil EC, TC, OM, MWD, clay and silt (Table 3), which indicates that an increase in any of these negatively correlated soil properties will decrease soil wind erodibility, leading to reduced amounts of aerosol produced by playa surfaces. The only positive correlation is to the sand fraction, implying that an increase in sand fraction of soils accelerates wind erosion.

**Table 3.** Pearson correlation coefficients between soil erodible fraction and physicochemical properties of playa surfaces.

| | EC | pH | $Ca^{2+}+Mg^{2+}$ | $Na^+$ | $Cl^-$ | TC | OM | MWD | Clay | Silt | Sand |
|---|---|---|---|---|---|---|---|---|---|---|---|
| EF | -0.45** | -0.06$^{ns}$ | -0.10$^{ns}$ | -0.08$^{ns}$ | -0.09$^{ns}$ | -0.39* | -0.39* | -0.77** | -0.55** | -0.51** | 0.64** |

Number of samples for analysis were 130. * and ** indicate significance at 5 % and 1 % probability levels, respectively. ns: not significant.

The size of soil aggregates is known to be an important and determining factor in controlling soil susceptibility to wind erosion. Studies have shown that soil wind erodibility is also affected by soil texture and organic matter (Motaghi et al., 2020). Soil organic matter and clay can improve soil aggregation by gathering soil particles together and binding them, leading to larger aggregates, which are more resilient against wind erosion (Charman and Murphy, 2000; Ghaderi and Ghodoosi, 2005; Colazo and Buschiazzo, 2010; Mahmoodabadi and Ahmadbeigi, 2013; Sirjani et al., 2019). The high EF of sa-sheets (Fig. 4b) can be explained by the high sand fraction along with low clay, OC and MWD, which makes this specific playa surface very susceptible to wind erosion (Virto et al., 2011; Motaghi et al., 2020). The absence of surficial crusts is another reason for high EF values. It has been demonstrated that in arid regions of USA, the destruction of surface soil accelerates wind erosion (Duniway et al., 2019).

The lowest EFs were observed in playa surfaces with high MWD, OM, clay and salt fraction, which includes clay flats (high clay, MWD, and OM) and salt crusts (high salt, MWD, clay, and OM) (Table 2, Fig. 4). Studies have shown that increase in soil OM and clay fraction increase the size and stability of soil aggregates through formation of macro-aggregates (>300 μm), which reduce the soil erodibility (Zobeck et al., 2013; Bronick and Lal, 2005; Shahabinejad et al., 2019). Sirjani et al. (2019) showed a significant negative correlation between wind erosion rate and soil properties including soil clay and OM contents. Shahabinejad et al. (2019) demonstrated that unlike clay and silt, sand fraction of the soils is directly proportional to wind erosion rates as the dominance of sand particles results in smaller aggregates due to their lower specific area and their surface charge. Due to the lack of water in arid and semiarid regions of the world, soils are less developed and therefore the sand fraction of the soils is high, making these parts of the world more susceptible to wind erosion (Osman, 2014). Thus, the size of the soil aggregates is a primary factor affecting the soil susceptibility to wind erosion (Charman and Murphy, 2000).

### 3.2 Dust source identification

Based on the physicochemical analyses presented in Sect. 3.1, locations with high susceptibility to wind erosion have been identified. In a next step, to confirm the relevance of these soils as dust sources, we compare surface collected soil with nearby air-sampled dust samples. In Table 4, the location of the soil and dust samples along with maximum wind speed and direction for the sampling day are presented. In Figs. A1 and A2 in Appendix, the wind rose plots for the sampling month (July 2020) and dry season (July–October 2020) are shown. In order to correlate the dust samples with their supposed source regions, we use two indicators: (i) correlation between minerals in the dust and soil samples, and (ii) calculation of enrichment factors from the elemental composition analysis of the dust and source soil samples.





**Table 4.** Location and time of soil and corresponding dust sample collection.

| Sample site | | Longitude (E) | Latitude (N) | Location (wrt UL) | Sampling date | Meteorological station | Maximum speed wind (m.s⁻¹) | Maximum wind direction (from) |
|---|---|---|---|---|---|---|---|---|
| Salmas (Sa) | dust | 44º49'10" | 38º12'38" | North | 11ᵗʰ July-2020 | Salmas | 18 | Southeast |
| | soil | 45º01'17" | 38º13'27" | | | | | |
| Jabal (Jab) | dust | 45º01'34" | 37º51'44" | Northwest | 9ᵗʰ July-2020 | Kahriz | 12 | Southeast |
| | soil | 45º03'22" | 37º50'36" | | | | | |
| Merange (Mer) | dust | 45º09'14" | 37º47'35" | West | 13ᵗʰ July-2020 | Urmia | 7 | East |
| | soil | 45º07'47" | 37º49'24" | | | | | |
| Miandoab (MD) | dust | 45º58'04" | 37º03'58" | South | 6ᵗʰ July-2020 | Miandoab | 12 | North |
| | soil | 45º46'44" | 37º06'49" | | | | | |

### 3.2.1 Mineralogical composition

Figure 6 shows the minerals identified in the soil and dust samples by the Rietveld refinement of X-ray diffractograms. In a first
step, we will analyze the composition of the soil samples, followed in second step, by a correlation between the composition of
soil and dust samples to deduce the sources of the dust samples.

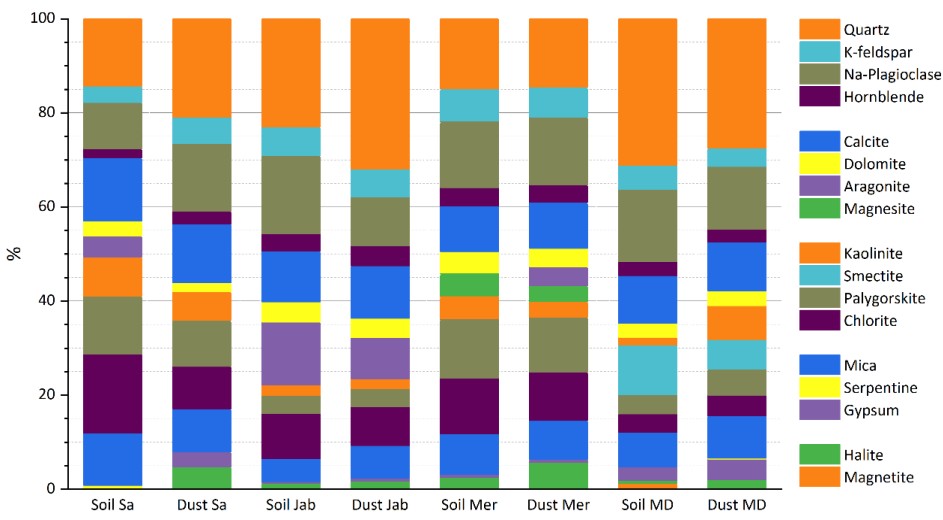

**Figure 6:** Mineralogical composition of natural soil and dust samples in % derived from Rietveld analysis of the X-ray diffraction (XRD) patterns.





In almost all soil samples, quartz was the dominant non-clay mineral varying between 14.3 % in Soil Sa and 31.1 % in Soil MD
(Fig. 6). Except for Soil Sa, silicate minerals (quartz, microcline, Na-Plagioclase and Hornblende) account for almost 50 % of the
soil mineral content. Carbonate minerals including calcite, dolomite, aragonite and magnesite range from 13.1 % (Soil MD) to
28.5 % (Soil Jab). Halite was observed in all investigated soil samples with the exception of Soil Sa, reflecting the low salt content
of this sample.

The highest phyllosilicate abundance (kaolinite + smectite + palygorskite + chlorite + mica + serpentine) was observed in Soil Sa
(49.4 %) and the lowest one in Soil Jab (20.5 %). As Soil Sa was from abandoned agricultural lands, which have recently been
exposed to secondary salinization, it had more suitable environmental conditions for soil development and therefore, it has a higher
clay content than other samples, where harsh environmental conditions such as high soil salinity, shallow groundwater table, and
anaerobic conditions limited soil development. Kaolinite, mica, palygorskite and chlorite were the common clay minerals observed
in the samples. Smectite was only present in Soil MD (Fig. 6). Motaghi et al. (2020) reported that illite, kaolinite, smectite and
chlorite are the common clay minerals of playa surfaces in the southeast of LU. They reasoned that these clay minerals are of
geologic rather than pedogenic origin as these soils experienced hardly any periods with favorable conditions for soil forming
processes. Similar results have been reported for western and southern parts of LU (Manafi, 2010; Hamzehpour et al., 2018). Illite,
vermiculite, kaolinite and mixed layer illite-smectite have been identified as dominant clay minerals near LU by Ahmady-Birgani
et al. (2015).

The dust samples exhibit a similar mineralogical composition as the soil samples with silicates, carbonates, and clay
minerals/phyllosilicates as the dominating species. This composition is in agreement with an intensive study carried out in
northwest (near Urmia) and southwest Iran, where Ahmady-Birgani et al. (2015) found that calcite, quartz, clay minerals and
gypsum are the main constituents of atmospheric particles. They have related the high calcite content of the samples to the
calcareous soils of the region. Several other studies from other parts of Iran or arid and semiarid regions throughout the world have
also reported that quartz and carbonate minerals are dominant constituents of atmospheric dust (Ganor et al., 2000; Jiries et al.,
2002; Zarasvandi et al., 2011; Díaz-Hernández et al., 2011; Al-Dabbas et al., 2012; Rashki et al., 2013b). Díaz-Hernández et al.
(2011) found that the predominant minerals in the aerosols observed in the southern Iberian Peninsula are quartz, calcite and
dolomite, which contribute 50 % to annual deposition followed by phyllosilicates (29.9 %), especially smectites, while halite
contributed 8 %.

Through long distance transport of mineral dust, the phyllosilicate content of the total mass of aerosol will increase because they
pertain to the fine aerosol fraction. However due to the regional origin of the collected dust in this study and their close distance
to the source, the non-clay fractions of the collected dust are mainly silicate and carbonate minerals, constituting more than 60 %
of the dust samples. The similarity between mineralogy of soil and dust in a region is an indicator of the regional origin of aerosols.
Therefore, dust mineralogy has been used for the identification of the different source regions worldwide (Krueger et al., 2004;
Yang et al., 2007; Rashki et al., 2013a; 2013b). In order to confirm the identified playa surfaces as dust sources, Pearson correlation
coefficients between minerals in soil and dust samples were calculated and are presented in Figs. 7 and 8. The overall lower
correlation among soils (0.69) than dusts (0.83) indicates that soils can be well discriminated from each other, while dusts are more
similar, due to the mixing of dust aerosols over the playa. The high correlation coefficients between minerals in soil and dust
samples from corresponding locations demonstrate that the selected soil samples are indeed main sources of the airborne dust,





evidencing the local contribution of the soils to the dust in the region. As an exception, Dust Sa showed weaker correlation with
       Soil Sa than the other soil/dust pairs (Fig. 7). It even correlates better with Soil Mer, Dust Mer, and Dust Jab, evidencing dust
       mixing in the northern part of LUP with dust coming from different parts of the playa.

| Mineral abundance | | Soils | | | | Dusts | | | | | | Averages of correlation coefficients between mineral abundance of | |
|---|---|---|---|---|---|---|---|---|---|---|---|---|---|
| | | Sa | Jab | Mer | MD | Sa | Jab | Mer | MD | | | | |
| Soils | Sa | | 0.68 | 0.86 | 0.52 | 0.83 | 0.66 | 0.84 | 0.63 | ↑ North | | all soils | 0.69 |
| | Jab | | | 0.69 | 0.76 | 0.78 | 0.92 | 0.79 | 0.77 | | | soils and southward originating dusts | 0.73 |
| | Mer | | | | 0.65 | 0.91 | 0.69 | 0.96 | 0.72 | South ↓ | | the group of soils and the group of dusts | 0.76 |
| | MD | | | | | 0.82 | 0.87 | 0.66 | 0.96 | | | soils and northward originating dusts | 0.79 |
| Dusts | Sa | | | | | | 0.83 | 0.92 | 0.90 | ↑ North | | all dusts | 0.83 |
| | Jab | | | | | | | 0.74 | 0.88 | | | dusts and soils at the same sampling site | 0.92 |
| | Mer | | | | | | | | 0.71 | South ↓ | | | |
| | MD | | | | | | | | | | | | |
| | | ← North   South → | | | | ← North   South → | | | | | | | |

**Figure 7:** Pearson correlation coefficients of minerals among soil and dust samples.

At each sampling site, halite is enriched in the dust compared with the corresponding soil sample. Dust Sa even consists of 4.7 %
       halite although this mineral is absent in Soil Sa. This indicates that halite in dust samples originates from other sources than covered
       by the investigated soil samples.



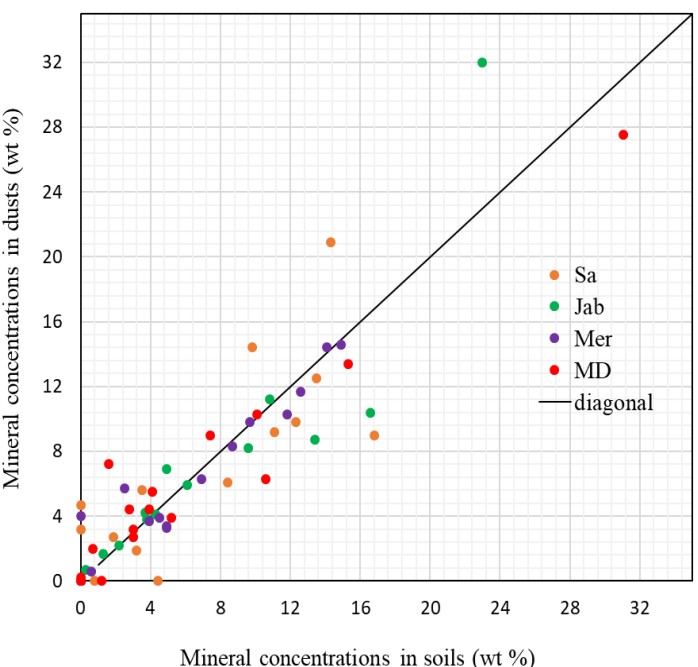

**Figure 8:** The correlation between mineral concentrations in soil and dust samples.


### 3.2.2 Elemental composition and enrichment factors in dust samples

The concentrations of the 32 elements determined by ICP-MS are presented in Table A1 in the Appendix, and the prevailing elements in each soil and dust sample are summarized in Table 5. Ca, Mg, Fe and Al were the most dominant elements in both soil and dust samples. The mean values for these elements are: Ca (120.3 g kg$^{-1}$); Fe (25.0 g kg$^{-1}$); Al (13.4 g kg$^{-1}$) and Mg (17.1 g kg$^{-1}$). Other abundant elements are Na, K, Si, while Sr, Ba, Zn, Cr, and Ni are present as trace elements. In a study carried out in the north and southeast of LU, Gholampour et al. (2017) found Fe, Al, P, Ti, and Mn as the dominant elements of the soil crust; Ca, P, Mg, Fe and K were present in deposited salts; and Fe, Al, Cu, Ti and P dominated in PM$_{10}$. In a recent study on rainwater chemical composition in the Lake Urmia basin, Ahmady-Birgani et al. (2020) explained the decrease in the concentrations of Na, K, Mg, and Ca$^+$ with increasing distance from LU with the influence of lakebed emissions on aerosol composition. Likewise, higher concentrations of marine elements (Na, K, Mg, and Ca) in dust samples in the present study compared to those of Gholampour et al. (2017) can be explained by the closer distance of the dust sampling sites to the erodible playa surfaces in our study.

**Table 5.** The most abandoned elements in soil and dust samples from LUP

| Prevailing elements |
| --- |





| Soil Sa | Ca > Mg > Fe > Al > Si > Na |
|---------|------------------------------|
| Dust Sa | Ca > Fe > Mg > Al > Na > Si > S > Ba |
| Soil Jab | Ca > Na > Mg > Fe > Al > Si > S > K > Sr |
| Dust Jab | Ca > Mg > Fe > Si > Al > Na > Sr > S |
| Soil Mer | Ca > Mg > Na > Fe > Al > K |
| Dust Mer | Ca > Fe > Al > Mg > Na > Si > S > Sr |
| Soil MD | Ca > Fe > Al > Na > Mg > K |
| Dust MD | Fe > Ca > Al > Na > Mg > Si > Zn > Ba |


**Figure 9:** Enrichment factors for elements in dust samples with respect to the corresponding soil samples (Table 4). Dashed lines at enrichment factors equal to 1, 5 and 10 are to guide the eye. Fe is used as reference element.

In order to validate the identified erodible playa surfaces as sources of dust in the studied areas, enrichment factors for each element
in dust samples with respect to the corresponding soil sample were calculated (see Sect. 2.2.6 for details) and are presented in Fig.
9. Most enrichment factors are equal or below one, indicating local contributions to the airborne dusts. Therefore, it can be
concluded that most elements in the dust samples including trace elements originate from the corresponding playa surfaces. This
is in accordance with Gholampour et al. (2017) who reported enrichment factors near unity for Al, Ti, Ca, P, Mn, K, and Si in
$PM_{10}$ in the north and southeast parts of LU. Behrooz et al. (2017) and MalAmiri et al. (2022) found soil crustal origin of Al, Fe
and Mg for airborne dust in southern Iran. Chen et al. (2007) showed by factorial analysis that elements such as Al, Si, Mg, K, Ca,
Fe in $PM_{10}$ originate from earth crust or soil.





In Dust Mer, enrichment factors for Ca, As, and Sr are higher than 5, and, with the exception of Dust Jab, the enrichment factor for Zn is higher than 10 indicating non-crustal emission sources. Indeed, several studies have suggested various anthropogenic sources contributing to the metal content of aerosols (Allen et al., 2001; Gatari et al., 2005; Wang et al., 2005; López et al., 2005; 470 Behrooz et al., 2017; MalAmiri et al., 2022). Industrial metallurgic processes, vehicle exhausts and fuel oil combustion can lead to increased concentrations of As, Cd, Cu, Ni and Zn (Bilos et al., 2001; Espinosa et al., 2001; Wang et al., 2005; Park and Dam, 2010; Gholampour et al., 2017). Chen et al. (2007) suggested that Zn and Pb in aerosols originate from industrial pollution while Cl and As may stem from combustion sources. Park and Dam (2010) assigned the concentrations of As, Cd, Pb, Se and Zn to road traffic and combustion sources. Similarly, MalAmiri et al. (2022) and Behrooz et al. (2017) reported that most of the trace elements 475 in airborne dusts from southern Iran are of anthropogenic sources such as fuel combustion, gas and petroleum drilling activities.

Soil and Dust Jab exhibit Sr concentrations of 2.8 g kg$^{-1}$ and 3.2 g kg$^{-1}$ (Table A1 in the Appendix), which is well above the average concentration in rocks (0.45 g kg$^{-1}$, Höllriegl, 2019). High strontium concentrations are found in basalt and carbonate rocks, where it can substitute calcium (Höllriegl, 2019). The predominance of carbonates in Soil Jab could at least partly explain the high concentration of Sr in this soil (Fig. 6). The enrichment factor of Sr for Dust Jab is close to one (Fig. 9), indicating that Sr in Dust 480 Jab mainly originates from Soil Jab. In Soil Mer, the strontium concentration is close to the average concentration in rocks (0.39 g kg$^{-1}$). However, in Dust Mer it is almost five times higher (1.9 g kg$^{-1}$) (Table A1 in the Appendix). As Dust Mer is collected close to Soil Jab (Fig. 1), it is likely that Sr in Dust Mer originates from Soil Jab, evidencing the mixing of dusts within LUP.

As discussed in Sect. 3.2.1, besides abandoned agricultural lands, erodible playa surfaces in the northern LUP also contribute to Dust Sa. Therefore, the high enrichment factor for Ba in this sample can be related to BaCl$_2$ emissions from LU lacustrine 485 sediments. Gholampour et al. (2015) demonstrated that BaCl$_2$ emitted from deposited sea salts can affect PM ionic characterization in Tasuj and Ajabshir regions located in northern and southeastern LU.

### 3.2.3 Physicochemical characteristics of soil and dust samples

A summary of the physicochemical properties of the soil and corresponding dust samples is presented in Table 6. Soil electrical conductivity (EC) varies among studied samples between 0.85 dS m$^{-1}$ in Soil Sa up to 43.8 dS m$^{-1}$ in Dust Jab with a mean value 490 of 23.56 dS m$^{-1}$ and standard deviation (SD) of 15.87 dS m$^{-1}$. Based on the criterion of 4 dS m$^{-1}$ which is the boundary between saline and non-saline soils, except for Soil Sa, all other studied samples are considered to be saline or even extremely saline (Weil and Brady, 2016).

**Table 6.** Physicochemical properties of the studied soil and dust samples.

| Sample | EC (dS m$^{-1}$) | TC (%) | OM (%) | pH | Particle size classes (< 2mm) | | | | | | |
| | | | | | Clay (%) | silt (%) | | | sand (%) | | |
| | | | | | | fine | medium | coarse | fine | medium | coarse |
| --- | --- | --- | --- | --- | --- | --- | --- | --- | --- | --- | --- |
| Soil Sa | 0.9 | 21.1 | 5.3 | 8.2 | 15.8 | 11.4 | 6.7 | 2.1 | 44.9 | 10.3 | 8.8 |
| Dust Sa | 27.5 | 14.4 | 2.6 | 7.5 | 5.7 | 25.4 | 17.8 | 20.0 | 20.1 | 8.1 | 6.9 |
| Soil Jab | 35.3 | 28.5 | 1.0 | 8.6 | 1.8 | 2.4 | 1.0 | 3.5 | 77.0 | 14.2 | 0 |
| Dust Jab | 43.9 | 43 | 1.3 | 8.3 | 1.1 | 1.2 | 0.4 | 0.4 | 51.0 | 43.7 | 2.3 |
| Soil Mer | 40 | 19.1 | 3.3 | 8.2 | 9.3 | 22.4 | 17.2 | 19.6 | 28.6 | 2.3 | 0.7 |



| | | | | | | | | | | | |
|---|---|---|---|---|---|---|---|---|---|---|---|
| Dust Mer | 19.9 | 17.1 | 2.9 | 8.2 | 3.0 | 5.2 | 2.6 | 2.4 | 21.6 | 61.6 | 3.7 |
| Soil MD | 16.3 | 13.1 | 1.2 | 8.1 | 3.0 | 7.5 | 2.9 | 2.7 | 60.6 | 12.4 | 10.9 |
| Dust MD | 4.9 | 13.5 | 1.7 | 8.0 | 3.9 | 8.6 | 3.4 | 3.6 | 69.0 | 9.7 | 1.0 |

EC: electrical conductivity; TC: total carbonates; OM: organic matter; clay: < 2 μm; silt: 2–50 μm, with fine (2–5 μm), medium (5–20 μm) and coarse (20–50 μm) fractions; sand: 50–2000 μm, with fine (50–200 μm), medium (200–500 μm), and coarse (500–2000 μm) fractions

Yet, there is only a very weak correlation (0.21) between EC and halite content of the soil and dust samples, indicating that also other soluble ions that are typical for lacustrine environments contribute to EC, including chlorides, sulfates, carbonates and bicarbonates of $Na^+$, $K^+$, $Ca^{2+}$ and $Mg^{2+}$. The prevalence of each of these ions varies from place to place, and also through time. In

case of LUP, Sharifi et al. (2018) reported that before desiccation of LU, the prevailing salts in brine of LU were NaCl, KCl and $MgSO_4$; while after its catastrophic water loss, salt composition has shifted to $NaSO_4$, $KSO_4$ and $MgCl_2$ and total dissolved salt content (TDS) has doubled.

Interestingly, Dust Sa exhibits very high salinity (EC = 27 dS m$^{-1}$) despite the low salinity of Soil Sa (EC = 0.9 dS m$^{-1}$). This discrepant salinity goes along with the absence of halite in Soil Sa compared with the considerable halite content in Dust Sa (4.7

%), and the strong enrichment of $BaCl_2$ in Dust Sa. Together with the generally lower correlation between Soil and Dust Sa compared with the correlations between the other soil/dust pairs, this again points to further, likely lacustrine sources contributing to Dust Sa.

Total carbonates (TC) of the soil samples vary between 13.1 % (Soil MD) and 28.5 % (Soil Jab), compared with values between 13.5 % (Dust MD) and 43 % (Dust Jab) for the dusts. TC from back titration are in excellent agreement with the sum of carbonates

(i.e. calcite + dolomite + aragonite + magnesite) from XRD for most soil/dust pairs, with the exception of Dust Jab, which exhibits a significantly higher TC concentration based on back titration than determined through mineralogical analysis (43 % versus 28.5%). This discrepancy could be due to the presence of non-crystallized carbonates (amorphous carbonates), which cannot be identified by XRD (Jiang et al., 2011).

Soil and Dust Jab exhibit the highest pH values of 8.6 and 8.3, respectively. For pH values up to 8.2, the main determining factor

is the presence of calcium-magnesium carbonates. Comparison of Table 6 with Fig. 6 shows that pH values are related to the total carbonate content of the samples. Since calcite and dolomite contents do not show significant variations among the samples, pH seems to be controlled by aragonite, which is a more soluble polymorph of calcium carbonate compared with calcite.

Mean organic matter of the samples was determined as 2.4 ± 1.4 %, with Soil Sa having the highest content with 5.3 %, which is indeed quite high for soils in the northwest of Iran, where OM is mostly lower than 3 % (Hamzehpour et al., 2018). As previously

discussed, Soil Sa is taken from an abandoned agricultural land and despite high OM content, which should prevent soil erodibility (Motaghi et al., 2020), this area has become a dust source due to destruction of soil aggregates and soil structure. The OM content of Dust Sa is 2.6 % (Table 6), which is higher than the mean soil OM in the area (Hamzehpour et al., 2018). Abandoning agricultural lands in arid and semiarid regions due to climate change and lack of water is becoming a serious problem. Aerosols originating from these regions have high OM contents and can eventually play an important role in ice nucleation activity of mineral dust

regionally or even globally (see following section).





In Table 6, particle size distributions of soil and dust samples after passing through the 2 mm sieve are presented. The classification of soil fractions in size classes is based on the criteria defined by the American Society for Testing and Material (ASTM, D-2487, 2000). Soil Sa has the highest clay fraction among the studied soil and dust samples (15.8 %). The decrease in clay fraction in the corresponding dust sample (5.7 %) further confirms that aggregation within the clay fraction reduces the susceptibility to wind erosion. Soil and Dust MD stand out as their particle size distribution exhibited a very high correlation (0.98). The fraction that is underrepresented in the dust is coarse sand (10.9 % in the soil compared with 1 % in the dust). This evidences the high susceptibility of the soil to wind erosion and the lack of other nearby sources. Soil and Dust Jab, which is classified as sa-sheets, had the lowest clay fraction (1.84 % and 1.1 %, respectively), which goes along with the highest erodible fraction of all investigated soils (89.7 %, Table 2). The particle size distribution of a soil is not only relevant for the susceptibility of soils to wind erosion, but also for long-range transport of dusts. Atmospheric aerosols mostly consist of silt sized (2–50 µm) and clay sized (< 2 µm) particles. Airborne particle concentrations diminish with distance from the source through time as material deposits by wet and dry processes. Aerosol particles smaller than 10 µm, which include clay (< 2 µm), fine silt (2–5 µm) and part of medium silt (5–20 µm) (Table 6), can be transported over long distances and may also contribute to cloud formation (Murray et al., 2012). Among the studied dust samples, Dust Sa had the largest share of particles < 2 µm and Dust Jab the lowest.

Overall, the mineralogical, elemental, and physicochemical correlations between soil and dust samples confirm that the investigated erodible playa surfaces from northwest to the south of LU, especially Soil Jab, are indeed major dust sources due to their unconsolidated substrates and scarce vegetation cover in most parts. Dust blowing over this region can raise huge amounts of particulate matter, and dust emitting from this area can become regionally or even globally important (Froyd et al., 2022). Indeed, it has been demonstrated that on a regional scale, the direct dust-climate feedback is enhanced by an order of magnitude near major dust source regions (Kok et al., 2018).

### 3.3 Ice nucleation ability of soil and dust samples

Figure 10 presents the DSC thermograms of the emulsion freezing experiments for the soil and dust samples for suspension concentrations of 2 and 5 wt %. The studied samples exhibit a homogeneous freezing peak between 236.2–236.9 K. Heterogeneous freezing occurred over a wide temperature range with freezing onset temperatures of 245.7–250.7 K for 5 wt % and 245.6–249.2 K for 2 wt % suspensions. For both suspension concentrations, Soil Sa exhibits the highest heterogeneous freezing onset temperature, $T_{het}$, and also the highest heterogeneously frozen fraction, $F_{het}$, with 250.7 K and 81.9 % for 5 wt % and 249.2 K and 78.0 % for 2 wt % suspensions, respectively. The lowest $T_{het}$ was measured for Dust Jab with 246.5 K for 5 wt % and 245.6 K for 2 wt % suspension concentrations. Moreover, Dust and Soil Jab exhibit the lowest $F_{het}$



**Figure 10:** DSC thermograms of the soil and corresponding dust samples in two concentrations; (a) 2 wt % concentration and (b) 5 wt % concentration. Thermograms have been normalized to the same area.



**Table 7.** Pearson correlation coefficient between mineral contents and $F_{het}$ and $T_{het}$ of natural soil and dust samples.

| Minerals | 5 wt % suspension | | 2 wt % suspension | |
|---|---|---|---|---|
| | $T_{het}$ | $F_{het}$ | $T_{het}$ | $F_{het}$ |
| Quartz | -0.44 | -0.47 | -0.28 | -0.18 |
| K-Feldspar (microcline) | -0.56 | -0.58 | -0.43 | -0.60 |
| Na-Plagioclase | -0.29 | -0.23 | -0.12 | -0.75* |
| Hornblende | -0.62 | -0.87** | -0.48 | -0.51 |
| Total silicates | -0.62 | -0.65 | -0.39 | -0.51 |
| Calcite | 0.25 | 0.52 | -0.04 | 0.28 |
| Dolomite | 0.02 | -0.57 | 0.07 | -0.13 |
| Aragonite | -0.26 | -0.60 | -0.43 | -0.54 |
| Magnesite | 0.10 | -0.04 | 0.21 | 0.08 |
| Total carbonates | -0.13 | -0.52 | -0.32 | -0.41 |
| Kaolinite | 0.53 | 0.72* | 0.38 | 0.65 |
| Smectite | 0.18 | 0.16 | 0.42 | 0.16 |
| Palygorskite | 0.44 | 0.59 | 0.32 | 0.43 |
| Chlorite | 0.49 | 0.41 | 0.22 | 0.30 |
| Total clay minerals | 0.78* | 0.87** | 0.65 | 0.70* |
| Mica | 0.58 | 0.83** | 0.47 | 0.83** |
| Serpentine | 0.80* | 0.69* | 0.56 | 0.70* |
| Total phyllosilicates | 0.76* | 0.88** | 0.63 | 0.74* |
| Gypsum | -0.20 | 0.15 | -0.02 | 0.06 |
| Halite | -0.55 | -0.19 | -0.55 | -0.34 |

Number of samples for analysis were 8. * and ** indicate significance at 5 % and 1 % probability levels, respectively.

In most of the 5 wt % suspensions, a second heterogeneous freezing peak with onset at around 244 K is present, which disappears in the 2 wt % suspensions, most probably because this peak arises from rather rare INPs, which become too scarce to give rise to 565 a distinct signal in the 2 wt % compared with the 5 wt % samples. Interestingly, the higher suspension concentration (5 wt %) goes along with higher freezing onset temperatures for all samples but not in all cases with a higher $F_{het}$, as some of the 2 wt % suspensions (Dust Jab, Dust MD, Soil Mer) exhibit higher $F_{het}$ than the corresponding 5 wt % suspensions. This indicates that there are interactions and impeding effects between sample components. Similar ranges of $T_{het}$ and $F_{het}$ of dust, and their corresponding





soil samples is in accordance with their similar mineralogical composition and physicochemical properties as discussed in Sects.
570     3.2.

The ability of mineral dust particles to nucleate ice depends on their mineralogical composition. Among the minerals present in the samples, quartz ($T_{het}$ = 247–251 K in emulsion freezing experiments), feldspars (specially K-feldspars with $T_{het}$ = 242–252 K) and clay minerals like kaolinite ($T_{het}$ = 239–242 K), illite ($T_{het}$ = 244–246 K) and montmorillonite ($T_{het}$ =239–247 K) have been found IN active (Pinti et al., 2012; Atkinson et al. 2013; Kumar et al., 2018; 2019a; 2019b; Harrison et al., 2019). Conversely,
minerals such as calcite, dolomite, and micas exhibit negligible IN activity (Kaufmann et al., 2016; Kumar et al., 2019b).

Correlation coefficients between $T_{het}$ and $F_{het}$ of soil and dust samples with their mineralogical composition are presented in Table 7. Based on these results, $F_{het}$ and $T_{het}$ correlate well with total phyllosilicates (kaolinite + smectite + palygorskite + smectite + mica + serpentine; $R$ = 0.63–0.88) and with the clay minerals (kaolinite + smectite + palygorskite + smectite; $R$ = 0.65–0.87).

Numerous studies showed that K-feldspars, especially microcline, nucleate ice at higher temperatures with higher activated
fractions than (Na-Ca)-feldspars (Peckhaus et al., 2016: Kumar et al., 2018; Kaufmann et al., 2016; Welti et al., 2019). Due to their high IN activity, K-feldspars have been considered key contributors to the IN activity of mineral dusts (Atkinson et al., 2013). Yet, quartz and microcline are anti-correlated with $F_{het}$ and $T_{het}$, although the freezing onset temperatures of the soil and dust samples fall in the typical range of these two minerals. The lack of correlation between K-feldspars could be due to their high sensitivity to the presence of ions and organic substances (Kumar et al., 2018; Whale et al., 2018: Yun, et al., 2020; 2021; Klumpp et al., 2022),
which constitute a relevant fraction of the investigated samples.

**Table 8.** Parson correlation coefficients between soil and dust physicochemical properties and $F_{het}$ and $T_{het}$ of soil and dust samples.

| Physicochemical properties | 5 wt % suspension | | 2 wt % suspension | |
|---|---|---|---|---|
| | $T_{het}$ | $F_{het}$ | $T_{het}$ | $F_{het}$ |
| EC (dS m$^{-1}$) | -0.68* | -0.75* | -0.57 | -0.45 |
| TC (%) | -0.3 | -0.46 | -0.28 | -0.53 |
| OM (%) | 0.79** | 0.77* | 0.44 | 0.74* |
| pH | -0.26 | -0.41 | 0.18 | -0.66 |
| Total clay fraction | 0.88** | 0.89** | 0.63 | 0.74* |
| < 10 μm | 0.46 | 0.58 | 0.33 | 0.5 |

EC: electrical conductivity; TC: total carbonates; OM: organic matter. Number of samples for analysis were 8. * and ** indicate significance at 5 % and 1 % probability levels, respectively.


The correlation of soil and dust physicochemical properties with IN activity (Table 8) shows a significantly positive correlation of $T_{het}$ and $F_{het}$ with total clay fraction in accordance with their positive correlation with clay minerals and phyllosilicates (Table 7), which are the major constituents of the clay size fraction. Moreover, there is a clearly negative correlation of $T_{het}$ and $F_{het}$ with EC, pointing to an inhibiting effect of ions on the IN activity of the samples. A negative effect of salinity could explain the absence of
a positive correlation between K-feldspar content and IN activity, as the presence of salts deteriorates the IN activity of feldspars (Kumar et al., 2018; 2019b). Finally, organic matter correlates positively with $T_{het}$ and $F_{het}$, supporting studies that have shown that



organics enhance the IN activity of soil dusts over the one of mineral dusts (Conen et al., 2011; O'Sullivan et al., 2014; Hill et al., 2016).

Taking into consideration the correlations presented in Table 8, the highest $T_{het}$ and $F_{het}$ among the studied soil and dust samples
for Soil Sa can be explained by their high OM fraction, but also by their high clay content and the high fraction of particles smaller than 10 µm (Fig. A3 in the Appendix). Moreover, the low salinity of Soil Sa could also contribute to the high IN activity. Although Dust Sa partly originates from Soil Sa (see Sect. 3.2), the higher soluble salts concentration (EC = 27.5 dS m$^{-1}$) together with the lower OM and lower total clay content (Table 6) compared to Soil Sa could explain its lower IN activity.

The comparatively low IN activity of Soil and Dust Jab among the studied samples, could be due to their high salt concentration
(EC), high total carbonate fraction (TC) and comparatively high pH together with the low clay content (Table 6) and particle fraction smaller than 10 µm (Fig. A3 in the Appendix).

**4 Conclusions**

Different playa surfaces have formed and are developing since desiccation of Lake Urmia in the northwest of Iran started. These barren lands have become new regional dust sources due to their extensive surface area, high salinity and low vegetation cover.
The study of the western LUP resulted in the identification of playa surfaces with different susceptibilities to wind erosion depending on their physicochemical properties and composition. The most resistant playa surfaces against wind erosion are salt crusts and clay flats while sand sheets and sandy salt crusts proved to be highly erodible. High sand fraction and low clay content along with low organic matter lead to small mean weight diameters of aggregates and are major factors increasing the wind erodible fraction of playa surfaces. Correlation of the mineralogical and elemental composition of airborne dust samples from the northwest
to the south of LU with highly erodible surfaces allowed identifying main sources of dust in the region. Mineralogical analysis determined quartz, carbonates and clay minerals (kaolinite, palygorskite and chlorite) as the prevailing minerals. The dominant elements are Ca, Fe, Al, Si, and Na and in some cases Ba, Sr, and Zn. Enrichment factors for most elements in the dust are close to one confirming their local crustal soil origin. Along with playa surfaces, agricultural lands, which are exposed to secondary salinization or water scarcity, had to be abandoned and have become dust sources. Ice nucleation activity of soil and dust samples
showed variations in IN activity depending on their mineralogical composition but also influenced by organic matter, salinity and pH. Specifically, IN activity showed positive correlations with organic matter and clay minerals, and was negatively correlated with high pH and high salinity, and, surprisingly, also with high K-feldspar and quartz content. To understand how the different components in the soil and dust samples contribute to the observed IN activity, further investigation of the samples through removal of components like OM, soluble salts, and carbonates, is required. The outcome of such an analysis is subject of Part 2 of this
series.

*Data availability.* The data presented in this publication will be submitted to the ETHZ data repository soon.

*Author contributions*. NH conducted the experiments and field works. SP contributed to field works. KK provided equipment and facilities for lab experiments. NH, CM, and TP contributed to the planning and interpretation of the experiments. NH prepared the manuscript with contributions from CM.

*Competing interests.* The authors declare that they have no conflict of interest.

*Acknowledgements.* We thank Michael Plötze, for XRD measurements; Lenny Winkel for ICP analysis; Marco Griepentrog for milling; Ulrich Krieger and Uwe Weers for support in the laboratory. We thank Daniel Wasner for providing size distribution



measurements (Laser diffraction particle sizer) (all ETH). We also thank Natural resources agency of West Azerbaijan, Iran for their contribution in dust collection.

*Financial support.* NH has been supported by visiting professors grant at ETH. KK has been supported by the Swiss National Foundation (project number 200021_175716).

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

**Appendix**

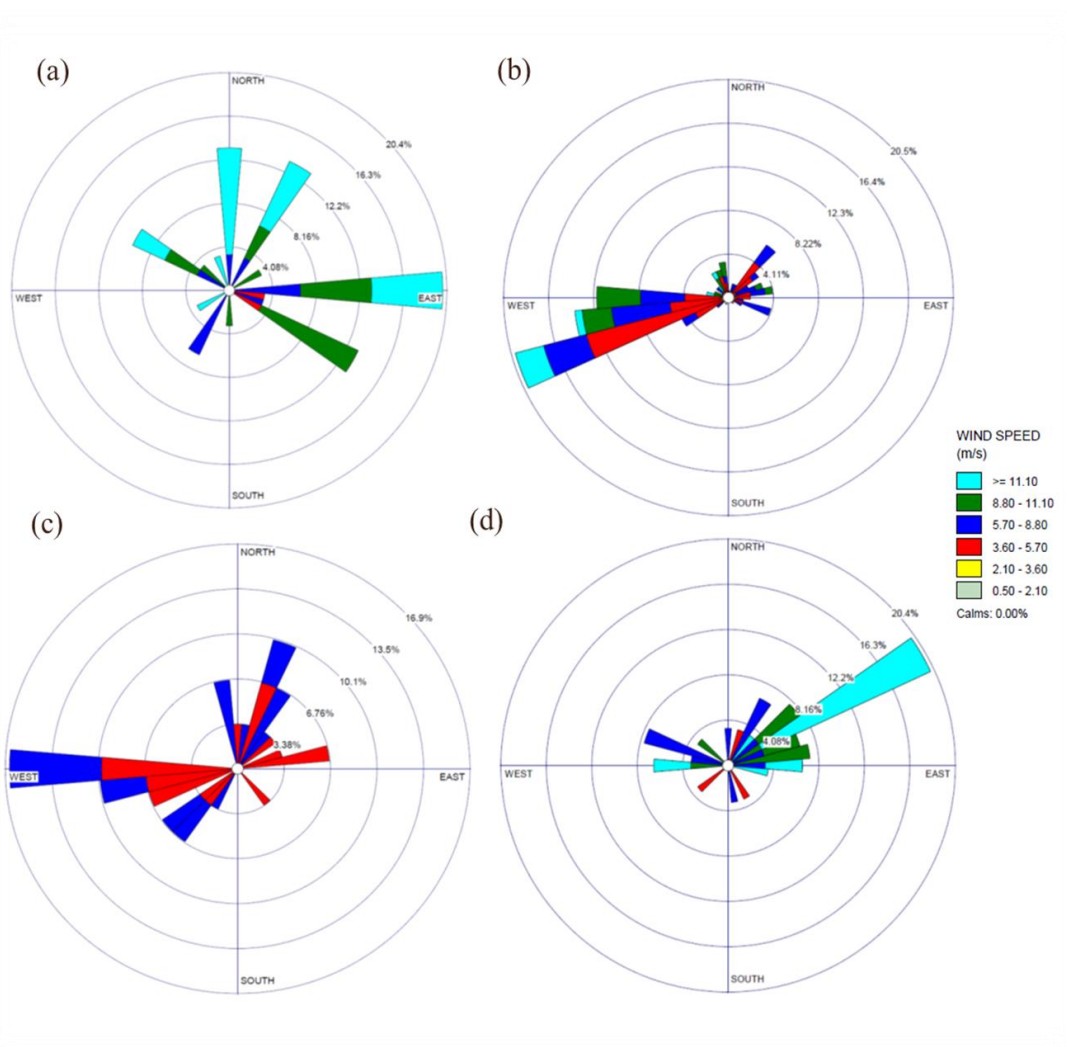

**Figure A1:** Wind rose plots for the maximum wind speeds and directions during sampling month July 2020 measured at the closest meteorological stations to the sampling locations. (a) Salmas station (west of Salmas soil sampling location); (b) Kahriz station (west of Jabal sampling location); (c) Urmia station (west of Merange sampling location); (d) Miandoab station (southeast of Miandoab sampling location). Lake Urmia is located east of Salmas, Kahriz, and Urmia stations and north of Miandoab station.




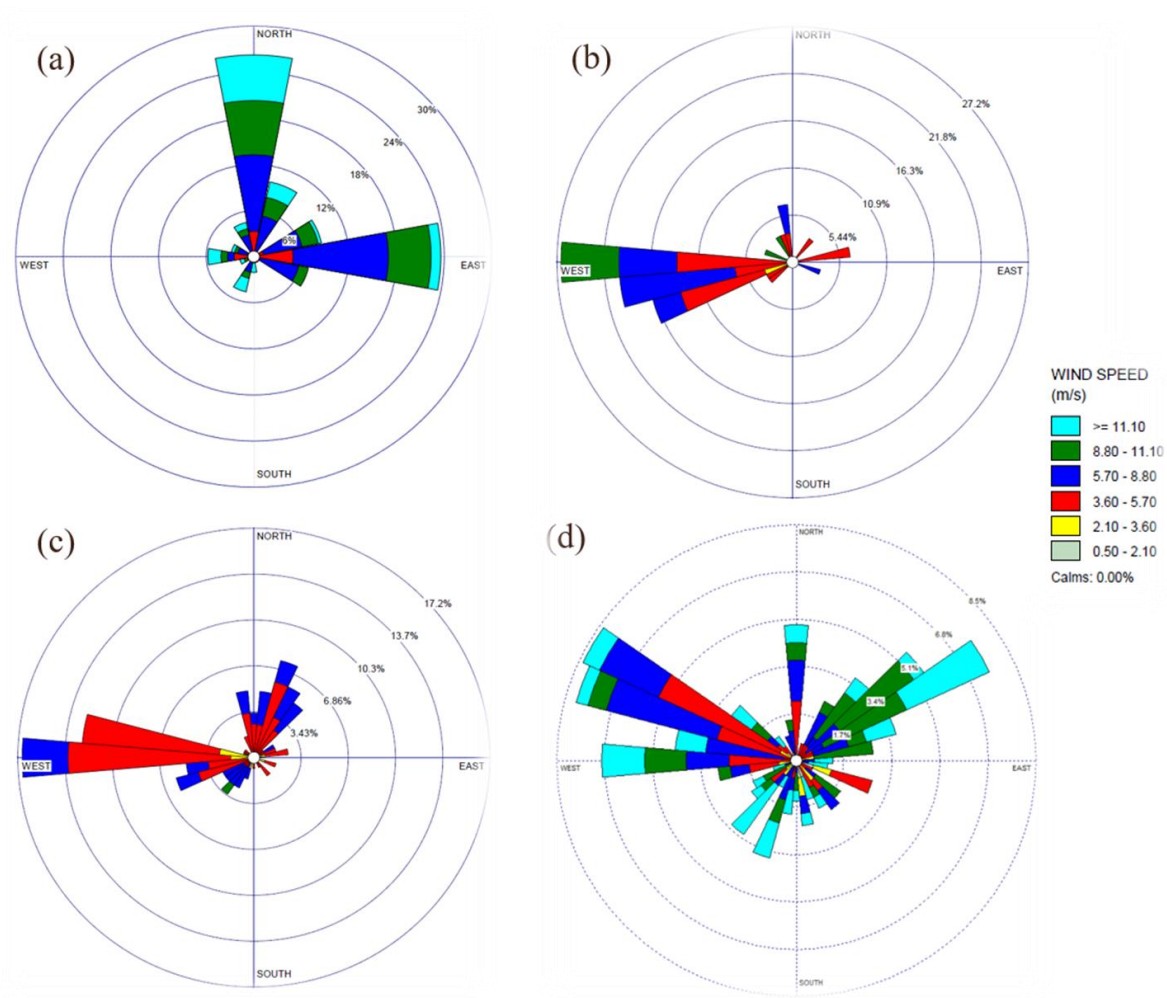


**Figure A2:** Wind rose plots for the maximum wind speeds and directions during dry months (June–October 2020) for the closest stations to the sampling locations. (a) Salmas station (Salmas sample); (b) Kahriz station (Jabal sample); (c) Urmia station (Merange sample); (d) Miandoab station (Miandoab sample).








**Figure A3:** Particle size distribution analysis of soil and dust fraction smaller than 63µm used for DSC analysis.



**Table A1:** Elemental constituents of soil and dust samples from highly erodible surfaces of Urmia Playa Lake.

| Elements (mgkg⁻¹) | Soil Sa | | Dust Sa | | Soil Jab | | Dust Jab | | Soil Mer | | Dust Mer | | Soil MD | | Dust MD | |
|---|---|---|---|---|---|---|---|---|---|---|---|---|---|---|---|---|
| | Average | StDev | Average | StDev | Average | StDev | Average | StDev | Average | StDev | Average | StDev | Average | StDev | Average | StDev |
| Li | 11.0 | 0.1 | 15.8 | 0.6 | 10.7 | 0.1 | 9.5 | 0.2 | 28.5 | 0.2 | 9.3 | 0.2 | 9.4 | 0.1 | 8.7 | 0.0 |
| Be | 0.4 | 0.0 | 0.7 | 0.0 | 0.3 | 0.0 | 0.2 | 0.0 | 0.7 | 0.0 | 0.4 | 0.1 | 0.6 | 0.0 | 0.5 | 0.1 |
| B | 83.2 | 0.2 | 133 | 7.3 | 209 | 4.1 | 178 | 9.0 | 388.3 | 18.1 | 130 | 3.2 | 75.7 | 0.2 | 85.5 | 1.1 |
| Na | 2904 | 54.1 | 6917 | 313 | 16974 | 226 | 5195 | 141 | 25601 | 1.1 | 8011 | 49.4 | 9863 | 84.6 | 9710 | 1.8 |
| Mg | 16781 | 116 | 23334 | 873 | 13160 | 141 | 10718 | 254 | 45817 | 210 | 9401 | 125.3 | 9017 | 68.5 | 8249 | 6.7 |
| Al | 12180 | 164 | 19034 | 136 | 7384 | 277 | 6130 | 1.0 | 163612 | 336 | 9749 | 99.4 | 19162 | 35.2 | 17076 | 124 |
| Si | 5214 | 626 | 4918 | 81.9 | 5880 | 423 | 6920 | 655 | 5192 | 189 | 7042 | 522 | 5322 | 106 | 5230 | 463 |
| P | 371 | 36.5 | 649 | 64.0 | 314.4 | 15.4 | 230 | 10.7 | 824 | 67 | 247 | 19.7 | 575 | 7.7 | 413 | 50 |
| K | 6444 | 134 | 8372 | 271 | 4107 | 28.6 | 3274 | 107 | 8319 | 61 | 5683 | 33.6 | 7590 | 33.8 | 6926 | 49 |
| Ca | 90516 | 483 | 75573 | 1107 | 228440 | 76.9 | 268537 | 4625 | 48479 | 608 | 168590 | 1557 | 34899 | 619 | 47502 | 535 |
| S | 895 | 44.4 | 3444 | 5.0 | 5374 | 316 | 3289 | 35 | 2240 | 132 | 2026 | 72.7 | 3833 | 84.7 | 4734 | 174 |
| V | 39.2 | 0.3 | 62.8 | 4.0 | 26.3 | 0.1 | 21.2 | 0.4 | 57.2 | 0.6 | 30.5 | 0.5 | 68.8 | 0.6 | 57.2 | 0.2 |
| Cr | 121 | 0.1 | 201 | 8.4 | 29.4 | 0.2 | 39.2 | 2.0 | 50.2 | 0.9 | 52.1 | 2.3 | 47.9 | 0.8 | 57.3 | 0.6 |
| Mn | 457 | 2.3 | 529 | 15.1 | 281 | 1.8 | 198 | 4.0 | 458 | 5.1 | 203 | 3.2 | 482 | 5.8 | 499 | 4.6 |
| Fe | 15319 | 195 | 31677 | 65.0 | 9346 | 15.4 | 7901 | 34 | 20664 | 164 | 12124 | 11.8 | 22510 | 326 | 80157 | 1382.7 |
| Co | 74.3 | 0.3 | 55.2 | 0.3 | 42.7 | 0.4 | 68.7 | 1.9 | 294 | 1.2 | 69.2 | 0.6 | 99.4 | 0.2 | 93.3 | 0.1 |
| Ni | 99.8 | 0.8 | 146.6 | 5.5 | 24.5 | 0.4 | 27.6 | 1.6 | 40.6 | 0.8 | 63.7 | 1.7 | 34.0 | 0.6 | 46.0 | 0.6 |
| Cu | 7.7 | 0.0 | 19.5 | 0.8 | 7.1 | 0.0 | 6.2 | 0.1 | 25.6 | 4.9 | 6.5 | 0.2 | 14.3 | 0.4 | 18.9 | 0.2 |
| Zn | 31.2 | 4.5 | 696 | 22.8 | 23.5 | 1.5 | 32.0 | 1.5 | 55.6 | 4.9 | 370 | 5.2 | 57.3 | 11.7 | 2708 | 40.0 |
| Ga | 5.1 | 0.0 | 7.6 | 0.3 | 3.2 | 0.1 | 2.4 | 0.1 | 7.6 | 0.1 | 4.6 | 0.1 | 8.1 | 0.1 | 8.3 | 0.1 |
| As | 13.9 | 0.5 | 17.3 | 0.5 | 19.8 | 1.1 | 18.5 | 0.3 | 8.5 | 0.3 | 30.2 | 1.1 | 16.1 | 0.3 | 15.9 | 0.3 |
| Se | 0.4 | 0.1 | n.a. | n.a. | 0.4 | 0.0 | n.a. | n.a. | 0.2 | 0.1 | 0.2 | 0.2 | 0.1 | 0.1 | 0.6 | 0.4 |
| Rb | 25.4 | 0.0 | 36.5 | 1.1 | 16.3 | 0.0 | 12.9 | 0.2 | 37.4 | 0.3 | 23.4 | 0.3 | 32.7 | 0.4 | 29.4 | 0.2 |
| Sr | 709 | 2.0 | 712 | 16.8 | 2796 | 28 | 3247 | 25 | 418 | 13 | 1919 | 0.2 | 179 | 3.9 | 393 | 13.9 |