# Peer review of "The Urmia Playa as a source of airborne dust and ice nucleating particles – Part 1: Correlation between soils and airborne samples"

_Atmospheric Chemistry and Physics, 2022_

## Author Response (AR1)

**We thank Reviewer 1 for his/her thoughtful comments. We reproduce the reviewer's comments in black and our responses in blue. Line numbers refer to the revised manuscript.**

The authors collected soil and dust samples in the Urmia Playa area in northwest of Iran. Physicochemical properties of soil and dust samples are analyzed. Mineralogical composition, elemental composition, and enrichment factors of dust samples are analyzed to identify the dust source. Moreover, the ice nucleation ability of soil and dust samples is also analyzed. The authors found that the IN activity positively depends on organic matter and clay minerals, while negatively depends on salinity, pH, K-feldspar, quartz, etc., which is very interesting.

The manuscript is well organized, the subject is relevant, the results are well presented and discussed, and, perhaps most importantly, the manuscript deals with an area where relevant observations are relatively rare. I believe the manuscript is suitable for publication after a minor revision.

Minor points:

I would like to see some discussions on what we learn from this study for the modeling community regarding the INP parameterization, in particular, for regional modeling over this area.

Reply: This is a very good question. Indeed, this region of the globe has gained little attention so far and we are not aware of regional climate modelling of it that includes INP properties. Also in global model studies like the recent study by Froyd et al. (2022), this region gained little attention in the discussion of the different emission sources. The present study shows that it is a regionally or even globally important source region for soil dust, yet, measurements performed by DSC cannot be directly converted to ice nucleation active site densities as required for input in models. We intend to do in future ice nucleation experiments with instruments that provide INAS. In addition, this region needs to be adequately represented as an emerging dust source region in models to yield meaningful results.

1. Why the dust sample locations are away from the soil sample locations? Please clarify.

   Reply: the collected dust samples have not been sampled far from the soil samples. They are collected at the meteorological stations in the nearby cities. The mean distance between soil and dust sampling locations is typically around 3 km. Like this, the local relevance of the soil sampling locations can be better judged compared with sampling soil and dust samples at exactly the same location. We collected samples in a reasonable distance from the soil samples to see:

   • If there are other dust sources participating in the composition of the airborne dust samples in the region.

   • If the dust coming from the playa surfaces affect the dust composition of the nearby cities.

2. The observed data and also post-processed data should be accessible even though during the review process.

   Reply: We have uploaded the observed and post-processed data to the ETH data repository.

3. P4, L130: during -> from.

   Reply: corrected

4. P4, L132: "potential evaporation value" -> "an annual potential evaporation value".

   Reply: corrected

5. P25, L563: In most 2 wt% suspension cases, the second heterogeneous freezing peaks are still there. Please clarify.

   Reply: Interestingly, the two investigated suspension concentrations yield INP densities so that the freezing signal in the 2 wt % samples is hardly lower or in some cases even larger than the one of the 5 wt % samples. We discuss this aspect in more detail in Part 2 of this paper series. In general, a small difference between the freezing signal of two different sample concentrations points to the presence of competing INPs with similar characteristic temperatures. In the Lake Urmia Playa samples, we have the additional effect of agents present in the samples that reduce or even inhibit the IN activity of INPs. We discuss this in Part 2.

6. The title: "source" -> "a source"

   Reply: corrected

**We thank Reviewer 2 for his/her comments. We reproduce the reviewer's comments in black and our responses in blue. Line numbers refer to the revised manuscript.**

This study analyses both soil samples and (airborne) dust samples collected in the drying region of the Lake Urmia (LU) in Iran. Besides for an analysis of parameters such as size distribution and composition, also ice activity was determined from differential calorimetry. While the characterization of the samples was well done, the use of differential calorimetry may not have given the full extent of information possible from off-line IN measurements. Particularly the freezing onset is not a very useful parameter, and it was neither defined how this nor the frozen fraction were defined and derived. Therefore, some unexpected results concerning the ice nucleation may arise from the applied methods, and this is difficult to judge from the present material.

Reply: We add definitions of freezing onset temperatures ($T_{het}$) and heterogeneously frozen fraction ($F_{het}$) to the revised manuscript. We think that these two parameters allow a good characterization of the IN activity of the investigated samples. The freezing temperatures that we obtain by differential scanning calorimetry (DSC) are comparable with those obtained by single-particle instruments such as continuous flow diffusion chambers. In our previous work (Pinti et al., 2012; Kaufmann et al., 2016; Kumar et al., 2018, 2019a; 2019b), we collected reference DSC thermograms of the most abundant minerals present in soils and dusts. Based on these, we can judge the IN activity of the samples. A more detailed analysis of the IN activity is given in Part 2 of this series.

Related to studies cited in the manuscript (lines 422-423), it also already has been known that similarity between mineralogy of soil and dust in a region is an indicator of the regional origin of aerosols, and dust mineralogy has been used for the identification of the different source regions worldwide. Therefore, the here presented study corroborates these findings but does not add much to it. It is also said clearly, that (line 520) "Abandoning agricultural lands in arid and semiarid regions due to climate change and lack of water is becoming a serious problem." This is an important statement which could have been stressed more.

Reply: One aim of this study was to determine the erodibility of the different playa surface types of LUP to identify the most erodible ones. In Sect. 3.1.3, we investigate the soil erodibility of 130 samples taken from the different types of playa surfaces and correlate them with other soil properties. Based on these analyses, we selected four soil and corresponding dust samples for more detailed analysis including IN activity. Previous studies that analyzed the IN activity of soil samples did not investigate (Tobo et al., 2014; Hill et al., 2016; Paramonov et al., 2018) nor even discuss the potential of their soil samples to become airborne (Conen et al., 2011; O'Sullivan et al., 2014).

We discuss in the introduction (lines 35–87) the problem of drying lakes in Iran that become new dust sources and give rise to saline dust storms. We also reference the relevant literature.

However, overall, the arising impression is, that while this study is a thorough characterization of dust in the region of LU, it does not extend far beyond that. As such, I advise the editor and the authors to discuss if this work is rather a measurement report and not a research article.

Reply: The relevance of this study needs to be judged together with Part 2 of this series, where we investigate the IN activity of the samples in more detail by selectively removing soluble

salts, carbonates, and organic matter. Mergin the two parts to one article would have led to a very long manuscript. Moreover, we think that the problem of drying-out lakes in arid and semiarid regions is a huge problem that deserves special attention. This paper highlights the Lake Urmia Playa as an emerging source region for large dust emissions. As we write on lines 54–56 of the introduction, there are more such lakes in Iran. We hope that with this study, the emergence of drying lakebeds in this region of the world will gain more attention when global inventories of dust source regions are updated, and that it motivates more research.

Besides for the above voiced criticism concerning the ice nucleation measurements, the study overall is a thorough study which merits publication, once the below addressed (smaller) issues will have been tackled.

Reply: Thank you for the valuation of the work we have invested in this study.

Specific comments:

Lines 27-28: Onset-temperatures are not very telling for judging an IN activity, and the onset temperatures you report here are rather low, compared to data from a range of other studies using filter samples and PCR-tray based off-line data evaluation (where an "onset" this is often found above ~260K (but typically not reported)). Therefore, claiming "high potential of dust blown from Urmia playa surfaces to affect cloud properties and precipitation" is exaggerative. Moreover, atmospheric INP number concentrations would be interesting, in this context, and it is not clear if they can be derived from your measurments.

(Publications I refer to here, to name only a few, are: Schneider et al. (2021), Testa et al. (2021), Gong et al. (2019) and even McCluskey et al. (2018) in clean marine regions.)

Reply: The temperature range where freezing occurs strongly depends on sample volume and concentration, and is therefore method and instrument specific. The publications that the reviewer mentions investigated filter-collected samples as they report atmospheric INP number concentrations per liter of air. As our samples are ground-collected, we cannot evaluate this quantity. Atmospheric dust concentrations are highly variable and strongly depend on local wind speed especially when the source is nearby. In addition, it is not possible to obtain INP number concentrations reliably through emulsion freezing experiments by DSC. The emphasis of our study is in correlating the IN activity of the samples with their composition. We compare the IN activity of the LUP samples among themselves and correlate them with composition. We do not just use the onset freezing temperature as the measure of IN activity but also the heterogeneously frozen fraction. With these two parameters, we cover the abundance of average INPs. We agree that further studies that investigate the IN activity of LUP samples more quantitatively, e.g. in terms of ice nucleation active site densities would be very interesting.

Line 164 ff: It did not become clear if the particle size distribution was only measured for soil samples or also for dust samples.

Reply: We did it for both types of samples. In line 166, we have mentioned "a Laser Diffraction Size Analyzer (LDSA; model LS 13320) was used to determine PSD between 0.4 μm to 2000 μm for the collected air-dried soil and dust samples". We use this information in Table 2 to divide the samples into clay, silt, and sand fraction.

Line 188: Please add if the surface collected soils that were compared with nearby air-sampled dust samples were among the original 130 samples mentioned in section 2.1.1? Or how was the

location for the selection of these soil dust samples chosen? Please also add if the collection was done at the same time as the dust sample.

Reply: In section 2.2.2 (lines 180–187) we describe the locations of the soil and dust samples. Two of the samples (Jabal and Merang) are among the 130 soil samples. The other two sampling locations were chosen according to the identification of major wind erodible surfaces of LUP from previous studies. To make this clearer, we specify on line 182:

"The soil samples Jabal (Jab) and Merange (Mer) were selected among the 130 samples collected as described in Sect. 2.1.1., and stem from two highly erodible playa surfaces identified in the western part of the LUP, namely Sa-sheets (sand sheets) and Sa-SC (salt-crusts) (Figs. 2b and 2c)."

We indicate the sampling dates in Table 1. We did the sampling for soil and dust samples the same day. To make this clear we add on line 394 of the revised manuscript:

"Note that we collected soil and the corresponding dust samples the same day."

Lines 251-252: More description is needed on the parameters introduced here – the reader should not have to look up another publication to obtain at least the needed basic information. Specifically: How was T_het determined, and how was F_het determined and what does it express? F_het often is a temperature dependent variable. Is it, in your case? Did you count separate droplets and frozen droplets and determined F_het from these? Or did you use the area under the thermograms? Could you add a figure showing an example, or showing F_het for all samples?

Reply: We have extended the description of the ice nucleation experiments starting on line 251:

"DSC thermograms of emulsion freezing experiments typically exhibit a homogeneous freezing peak arising from droplets that either are devoid of INPs (mostly droplets with diameters < 2 μm) or contain particles that are ice-inactive, and a heterogeneous freezing signal induced by INPs. Typical droplet size distributions of emulsions used in DSC experiments are depicted in Marcolli et al. (2007), Pinti et al. (2012), and Kaufmann et al. (2016). The onset of the heterogeneous freezing peak ($T_{het}$) and the heterogeneously frozen fraction ($F_{het}$) were evaluated as measures of the IN efficiency of the samples. Onsets were determined as the intersection of the tangent drawn at the point of greatest slope with the extrapolated baseline (see Fig. 1 of Kumar et al., 2018). $F_{het}$ was evaluated in the time domain of the thermograms as the ratio of the heterogeneous freezing signal to the total freezing signal (see Kumar et al., 2018a for details). Narrow sharp peaks in the thermogram, so-called spikes, arise from the freezing of large droplets with diameters of up to 500 μm and were excluded from the analysis. To test the stability of the emulsions, some samples were subjected to three freezing cycles following the procedure introduced by Marcolli et al. (2007) with a first and third cycle performed at a cooling rate of 10 K/min as control cycles. Emulsions were freshly prepared before each experiment. Every experiment was repeated at least once with a freshly prepared suspension. The average precision in $T_{het}$ is ±0.2 K. Uncertainties in $F_{het}$ are on average ±0.02, but may be much larger when heterogeneous freezing signals are weak or overlap (forming a shoulder) with the homogeneous freezing signal."

Lines 319-321: These two sentences are a bit contradictory, as you say in the first sentences, that soil organic matter is present in relatively low amounts, but then you say that these are

typical values. So what does "relatively low amount" refer to? (The two publications you cite here give values in a similar range.)

Reply: due to low rainfall, high temperatures, and mostly because of improper land management practices in most parts of Iran, overall organic matter content of the soils are low compared with other regions on the globe. Northwest Iran has lower mean annual temperature and higher rainfall in comparison to the central Iran. However, organic matter content is still within the range reported for the soils from different parts of the country. To make this clear, we revise starting from line 331:

"Compared with other regions on the globe, soil organic matter is only present in relatively low amounts and varies between 0.5 % and 6.0 % (Table 2). Except for beach sand with 6.0 %, these are typical values for soils in the western LU region, where OM is mostly below 3 % (Hamzehpour et al., 2018)."

Line 426 ff: Is it fair to assume that mixing of dusts while they are airborne explains your finding that there is an overall lower correlation coefficient between soils than dusts? If so, maybe add this line of thought to the text.

Reply: Assuming that the dust originates from local sources, this is the only explanation for the higher correlation between dusts than soils that we can think of. Long-range transport of the dust samples can be excluded considering the high sand fraction of the dust samples, which can only be transported over short distances. We make this line of thought clearer by revising starting from line 439:

"The overall lower correlation among soils (0.69) than dusts (0.83) indicates that soils can be well discriminated from each other, while dusts are more similar. The high correlation between minerals in soil and dust samples from corresponding locations demonstrates that the selected soil samples are indeed main sources of airborne dust, evidencing the local contribution of the soils to the dust in the region. Given that the dust source is local, the higher correlation between dusts than soils evidences the mixing of dust over the playa."

Line 551: Again: How is F_het defined? It is difficult to judge your results if it not clear how this parameter was derived.

Reply: We have added the definition of $F_{het}$ to Sect. 2.3.

Figure 10: What is indicated by the temperatures given in the plots? Are these onset temperatures? Again: How are they derived, anyway?

Reply: Thank you for pointing this out. We add the following information to the figure caption:

"Numbers on the left display $F_{het}$. The temperatures $T_{het}$ are given at the onsets of the heterogeneous freezing peaks,"

Line 565 ff: Could these observations also originate from peculiarities of the DSC-technique? Is there a chance to repeat these measurements with other off-line INP analysis techniques of close by befriended groups? This is not too much work and could clarify if you are really onto something here.

Reply: We indeed intend to re-measure the IN activity of some of the LUP samples with other types of instruments. In frequently used droplet freezing assays, the second freezing peak at around 244 K should appear as uneven increase of frozen fraction with decreasing temperature. Yet, in this type of setup, droplet volumes are in the microliter range, so that the number of particles within each droplet is much higher than in emulsion droplets. This shifts the frozen fraction curve to higher temperatures compared with emulsion freezing experiment. Moreover, due to the much larger sample volume in droplet freezing assays, ubiquitous impurities induce freezing of "pure" water already around 250 K, so that features at 244 K are not accessible. We would need sample volumes in the picoliter range as covered e.g. by microfluidic devices to obtain frozen fractions at this temperature. Such measurements would go beyond the scope of this paper.

And, as mentioned above, onset temperatures are not a very informative parameter, anyway, and F_het was not defined. Much could be gained by additional measurements.

Reply: $F_{het}$ together with $T_{het}$ are both given in Fig. 10. Moreover, we have added the definition of these parameters to Sect. 2.3. DSC thermograms of the LUP samples can be compared with reference DSC thermograms of most atmospherically relevant mineral particle types that we have collected over the last years (Pinti et al., 2012; Kaufmann et al., 2016; Kumar et al., 2018, 2019a; 2019b). We present a more in-depth analysis and discussion of the freezing signals in Part 2 of this series.

Lines 583-585: If organic substances and ions would mask the ice nucleation by K-feldspar and quartz, as you say, higher onset temperatures may be expected. Have you tried if heating the samples changes the results on ice nucleation? If you did, but this is part of the second paper, maybe at least point towards this here.

Reply: We have investigated the effect of removing soluble ions and organic matter (through $H_2O_2$ digestion) on the IN activity of the samples. We find that $T_{het}$ in most samples decreases after removing organic matter, and increases when removing soluble ions. The effect of such treatments is indeed the subject of Part 2, where we conclude that organic matter determines the onset freezing temperatures of most samples.

Following the suggestion of the reviewer, we revise the manuscript starting from line 599:

"In Part 2 of this series we show that the onset freezing temperatures of most samples decreases when organic matter is removed, and increases when soluble salts are removed. The negative effect of soluble ions on the IN activity of feldspars has been shown in several studies (Kumar et al., 2018; Whale et al., 2018; Yun et al., 2020)."

Lines 599-601: Is there a chance to determine the particle surface area of both soil and dust samples to tackle the surface area dependence of ice nucleation and therewith to make these two groups of samples comparable?

Reply: We analyze the samples with respect to composition and size distribution, which we split in clay, silt, and sand fraction. We think that this characterization is more meaningful than the total surface area of such mixed samples. Our analysis shows that dust and soil samples are very well comparable.

Minor and editorial comments:

Line 263: In your manuscript it is sometimes "sa-sheet", sometimes "Sa-sheet". As this is some kind of a parameter, it should not change but be consistently the same at all occurrences. As you capitalize most of the other abbreviations, it would be best to also do it for that one.

Reply: We corrected to Sa-sheets throughout the manuscript.

Table 1: It is confusing that you mix long and abbreviated sample names, i.e., giving both for some and either one or the other for others. Preferentially, both would be given here for all samples, so one could refer to this table and would not go back to the text where this is defined if one wanted to look that up again.

Reply: We added the full names to this table when they were missing and give their abbreviations in brackets as suggested by the reviewer.

Also: Fan delta is not included in Table 1. It is also not included in Table 2. Why is that? Is it, because (line 291) "No soil samples were taken from these locations due to waterlogging."? If so, why is it mentioned and included at all, in your text? Explain this when you introduce "Fan delta" in the text.

Reply: Identifying fan deltas both in the field and also from satellite images is easy and is of high precision. However, as we did not take soil samples from these regions due to their inaccessibility at the time of the sampling, we cannot comment on their properties. Therefore, they are excluded from Table 2.

Figure 3: According to Fig. 1, black circles should be the dust sampling sites. Here it says it's the white circles?

Reply: You are right. We corrected to "The dust and soil sampling sites (black and white circles, respectively)."

Lines 526-528: Please add already here that the size classes for clay, silt and sand can be found in Table 6.

Reply: Corrected as suggested.

Table 6: "Clay" is capitalized, "silt" and "sand" are not. Unify.

Reply: Corrected to "Clay, Silt and Sand".

Line 543: As you cite Froyd et al., 2022 here, make clear that they refer to the Middle Eastern region as a whole, not to the LU region in particular.

Reply: We follow the reviewer and revise to (line 559): "Dust emissions from the Middle Eastern region can become regionally or even globally important (Froyd et al., 2022)."

Line 544 ff: "… it has been demonstrated that on a regional scale, the direct dust-climate feedback is enhanced by an order of magnitude near major dust source regions (Kok et al., 2018)." This enhancement is compared to what? Please add.

Reply: We specify in the revised manuscript on line 560: "it has been demonstrated that on a regional scale, the direct dust-climate feedback is enhanced over its globally averaged value by an order of magnitude near major dust source regions (Kok et al., 2018)."

Literature:

Gong, X., H. Wex, T. Müller, A. Wiedensohler, K. Höhler, K. Kandler, N. Ma, B. Dietel, T. Schiebel, O. Möhler, and F. Stratmann (2019), Characterization of aerosol properties at Cyprus, focusing on cloud condensation nuclei and ice nucleating particles, Atmos. Chem. Phys., 19, 10883-10900, doi:10.5194/acp-19-10883-2019.

McCluskey, C. S., J. Ovadnevaite, M. Rinaldi, J. Atkinson, F. Belosi, D. Ceburnis, S. Marullo, T. C. J. Hill, U. Lohmann, Z. A. Kanji, C. O'Dowd, S. M. Kreidenweis, and P. J. DeMott (2018), Marine and Terrestrial Organic Ice-Nucleating Particles in Pristine Marine to Continentally Influenced Northeast Atlantic Air Masses, J. Geophys. Res.-Atmos., 123(11), 6196-6212, doi:10.1029/2017jd028033.

Schneider, J., K. Hohler, P. Heikkila, J. Keskinen, B. Bertozzi, P. Bogert, T. Schorr, N. S. Umo, F. Vogel, Z. Brasseur, Y. S. Wu, S. Hakala, J. Duplissy, D. Moisseev, M. Kulmala, M. P. Adams, B. J. Murray, K. Korhonen, L. Q. Hao, E. S. Thomson, D. Castarede, T. Leisner, T. Petaja, and O. Mohler (2021), The seasonal cycle of ice-nucleating particles linked to the abundance of biogenic aerosol in boreal forests, Atmos. Chem. Phys., 21(5), 3899-3918, doi:10.5194/acp-21-3899-2021.

Testa, B., T. C. J. Hill, N. A. Marsden, K. R. Barry, C. C. Hume, Q. J. Bian, J. Uetake, H. Hare, R. J. Perkins, O. Mohler, S. M. Kreidenweis, and P. J. DeMott (2021), Ice Nucleating Particle Connections to Regional Argentinian Land Surface Emissions and Weather During the Cloud, Aerosol, and Complex Terrain Interactions Experiment, J. Geophys. Res.-Atmos., 126(23), doi:10.1029/2021jd035186.

Literature

Conen, F., Morris, C. E., Leifeld, J., Yakutin, M. V., and Alewell, C.: Biological residues define the ice nucleation properties of soil dust, Atmos. Chem. Phys., 11, 9643–9648, https://doi.org/10.5194/acp-11-9643-2011, 2011.

Hill, T. C. J., DeMott, P. J., Tobo, Y., Fröhlich-Nowoisky, J., Moffett, B. F., Franc, G. D., and Kreidenweis, S. M.: Sources of organic ice nucleating particles in soils, Atmos. Chem. Phys., 16, 7195–7211, https://doi.org/10.5194/acp-16-7195-2016, 2016.

Kaufmann, L., Marcolli, C., Hofer, J., Pinti, V., Hoyle, C. R., and Peter, T.: Ice nucleation efficiency of natural dust samples in the immersion mode, Atmos. Chem. Phys., 16, 11177–11206, https://doi.org/10.5194/acp-16-11177-2016, 2016.

Kumar, A.; Marcolli, C.; Luo, B., and Peter, T. Ice nucleation activity of silicates and aluminosilicates in pure water and aqueoussolutions−Part 1: The K-feldspar microcline. Atmos. Chem. Phys., 18, 7057−7079, https://doi.org/10.5194/acp-18- 7057-2018, 2018.

Kumar, A., Marcolli, C., and Peter, T.: Ice nucleation activity of silicates and aluminosilicates in pure water and aqueous solutions – Part 2: Quartz and amorphous silica, Atmos. Chem. Phys., 19, 6035–6058, https://doi.org/10.5194/acp-19-6035-2019, 2019a.

Kumar, A., Marcolli, C., and Peter, T.: Ice nucleation activity of silicates and aluminosilicates in pure water and aqueous solutions – Part 3: Aluminosilicates, Atmos. Chem. Phys., 19, 6059–6084, https://doi.org/10.5194/acp-19-6059-2019, 2019b.

Marcolli, C., Gedamke, S., Peter, T., and Zobrist, B.: Efficiency of immersion mode ice nucleation on surrogates of mineral dust, Atmos. Chem. Phys., 7, 5081–5091, doi:10.5194/acp-7-5081-2007, 2007.

O'Sullivan, D., Murray, B. J., Malkin, T. L., Whale, T. F., Umo, N. S., Atkinson, J. D., Price, H. C., Baustian, K. J., Browse, J., and Webb, M. E.: Ice nucleation by fertile soil dusts: relative importance of mineral and biogenic components, Atmos. Chem. Phys., 14, 1853–1867, https://doi.org/10.5194/acp-14-1853-2014, 2014.

Paramonov, M., David, R. O., Kretzschmar, R., and Kanji, Z. A.: A laboratory investigation of the ice nucleation efficiency of three types of mineral and soil dust, Atmos. Chem. Phys., 18, 16515–16536, https://doi.org/10.5194/acp-18-16515-2018, 2018.

Pinti, V., Marcolli, C., Zobrist, B., Hoyle, C. R., and Peter, T.: Ice nucleation efficiency of clay minerals in the immersion mode, Atmos. Chem. Phys., 12, 5859–5878, https://doi.org/10.5194/acp-12-5859-2012, 2012.

Tobo, Y., DeMott, P. J., Hill, T. C. J., Prenni, A. J., Swoboda-Colberg, N. G., Franc, G. D., and Kreidenweis, S. M.: Organic matter matters for ice nuclei of agricultural soil origin, Atmos. Chem. Phys., 14, 8521–8531, https://doi.org/10.5194/acp-14- 8521-2014, 2014.

---

## Author Response (AR2)

Suggestions for revision or reasons for rejection (will be published if the paper is accepted for final publication)

The authors responded to the questions raised by the reviewers and improved the (already good) manuscript such, that I now can recommend publication in ACP.

However, I still want to point out that this may be better suited as a measurement report instead of a research article. The answer to this point was "The relevance of this study needs to be judged together with Part 2 of this series", but that exactly underlines that likely more in-depth research is contained in Part 2. So this Part 1 could be a measurement report, laying out the basis, while then Part 2 could be an research article. But I won't insist on this and leave it up to the editor to decide this.

Reply: Authors thank reviewer for his/her valuable comments which have improved the manuscript.

Concerning the reviewer suggestion about the considering the manuscript as " measurement report" type, we should comment that Part 1 and Part 2 of our series have been conceived as research articles and include a discussion of the results, which exceeds a pure report of measurements. In Part 1, we determine different playa surfaces of Lake Urmia based on understanding the land evolution under harsh conditions, along with analysing the erodibility of soils. We also connect ice nucleation ability of dust samples to several soil properties from dust sources and finally we identify determining factors of ice nucleation ability of dust. We consider such a combined discussion as crucial, as both erodibility and ice- nucleation activity are required for a soil to be a source of ice-nucleating particles for cloud formation. Often ice-nucleating activities of soil samples are studied without considering the erodibility of the soils. Part 1 provides such a discussion. We therefore think that Part 1 fulfils the criteria for a research article."

There were a few typos and such, which you should check. When giving line numbers, I refer to the version with "tracked changes".

Line 22: I know that I asked for more precise naming of the samples, but using the abbreviation of one sample "(Sa-sheets)", and only that one, in the abstract is overdoing it. Please remove again.

Reply: we removed the Sa-sheets from the abstract.

Line 260: Please add (in case I get this right), that $T\_het$ and $F\_het$ are both concentration dependent and that these two parameters are used in your study to compare the IN activity among the studied dusts. That would have helped my understanding.

Reply: this sentence is added to the line 256: "As $T_{het}$ and $F_{het}$ are also affected by sample concentration, all samples are evaluated at 2 % and 5 % concentrations."

Table 1: Sand sheets are in this table twice, now (in the first column). Once certainly should be the mix of Sand sheets and Salt crust.

Reply: Thank you very much for your comment. I think while replacing Sa-sheets throughout the manuscript Sandy Salt Crust (Sa-SC) is replaced by Sand sheets. I corrected it in the Table 1.

Table 2: It seems you added an "s" to "…chemical" (it is now "chemicsal"), but that should be removed again.

Reply: I corrected the typo in Table 2.

Figure 10: The lowermost curve (Dust MD for the 2 wt% samples) likely should have been a yellow curve. Not that it matters, I just saw that.

Reply: that is correct. I changed the color of Dust MD to yellow at 2 wt%.